# DP-Nav: Dynamic Exploration Driven by Semantic Region Potential for Zero-shot Visual Navigation

## Abstract

Visual navigation requires the agent to autonomously navigate to a specified goal based on sequential visual perception. A key challenge is to achieve target localization and optimize the path simultaneously. However, most existing frontier-based methods rely on static navigation policies, which update the target frontiers at fixed time intervals to guide the agent's exploration. These approaches cannot dynamically assess potential regions encountered during navigation, thereby preventing timely policy adjustments. Moreover, the presence of multiple frontiers within the same region often leads to repeated exploration of identical regions, further exacerbating path redundancy and inefficiency. To address the above limitations, we propose DP-Nav, a novel dynamic navigation framework driven by the potential of semantic regions. Our approach first identifies distinct semantic regions from sequential visual perception and treats an independent semantic region as a policy unit. Furthermore, we introduce a Scoring-Screening Mechanism (SSM) that evaluates and filters these semantic regions based on their potential utility. Then SSM assigns exploration priorities to different regions, selecting the semantic region with the highest potential value for the agent's subsequent exploration. More significantly, we design a Dynamic Policy Trigger (DPT) module that enables on-demand activation of the SSM, allowing the agent to dynamically adapt its exploration policy in response to environmental changes and perceptual feedback, thereby addressing the rigidity of static policies. Extensive experiments on Object Goal Navigation, Text Goal Navigation, and Instance Image Goal Navigation across Gibson, HM3D, and MP3D datasets demonstrate that DP-Nav achieves SOTA performance and improves path efficiency by about $7\% \sim 17\%$.

## 1 Introduction

As the foundation task of Embodied Navigation Das et al. (2018); Majumdar et al. (2024), visual navigation enables the agent to autonomously explore unseen environments and locate specified goals, typical task variants include Object Goal Navigation (ObjectNav) Chaplot et al. (2020a); Yin et al. (2024), Text Goal Navigation (TextNav) Sun et al. (2024); Yin et al. (2025), and Instance Image Goal Navigation (InstanceNav) Krantz et al. (2023); Lei et al. (2024). Despite different goal specifications, they share the common challenge of guiding the agent to succeed in reaching the goal while ensuring an efficient navigation path.

To achieve this objective, many current methods Yu et al. (2023b); Kuang et al. (2024); Yin et al. (2025) adopt Frontier-based Exploration (FBE) Yamauchi (1997) policy. Central to these FBE methods Zhou et al. (2023); Yokoyama et al. (2024); Long et al. (2024) is evaluating the semantic relevance between frontiers and the specific goal, then selecting the frontier with the highest relevance for exploration. While proven effective, they have two significant limitations. First, Current navigation policies are mostly static. They update target frontiers at fixed time steps to guide agent exploration. This approach differs significantly from how humans make decisions Treisman & Gelade (1980). Humans adjust their search direction in real-time based on perception feedback and environmental changes. Existing methods Zhang et al. (2024); Yokoyama et al. (2024); Yin et al. (2025) cannot respond quickly to such information. As a result, the agent often misses promising semantic regions along the path. For example, when looking for a chair, an agent might pass by

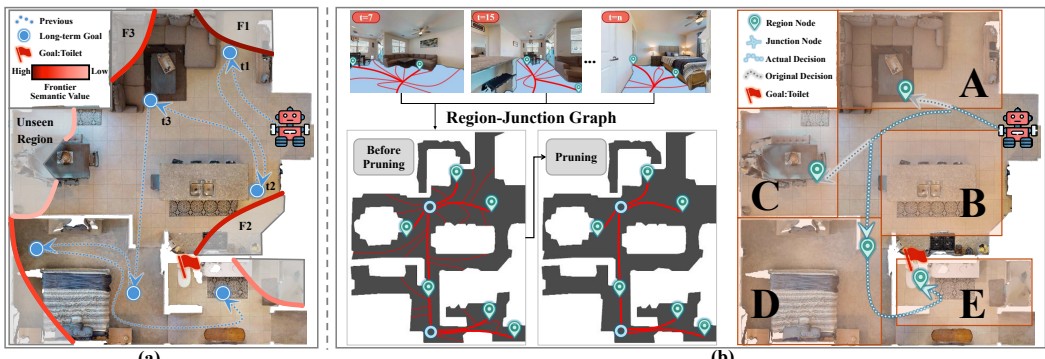

Figure 1: The previous agent (Subfigure **a**) first explores Frontier $F_1$ (Region A) at $t_1$, then diverts to Frontier $F_2$ (Region B) at $t_2$, and returns to Frontier $F_3$ (Region A) at $t_3$. This is because multiple frontiers in a semantic region lead to region fragmentation, resulting in cross-region backtracking and additional path redundancy. While our semantic region-aware exploration (Subfigure **b**) could prevent re-exploration of the same region and dynamically adjust the policy to optimize the path.

a room. It can quickly check whether the target is inside with a minimal path cost. If not, going back later would waste more travel distance. This weakness reduces overall exploration efficiency and increases path cost. Second, existing FBE methods treat individual frontiers as decision units. However, one semantic region often contains multiple frontiers. This may cause the agent to jump inconsistently between different semantic regions. For instance, it may revisit the same region multiple times through different frontiers, as illustrated in the Figure 1 (**a**). Such behavior leads to repeated coverage of the same region, which further reduces path efficiency.

To overcome the aforementioned limitations, this paper proposes DP-Nav—a dynamic exploration framework driven by the potential of semantic regions. The method begins by extracting traversable areas frame by frame from a real-time RGB-D sequence and recognizes semantic regions in the current frame using a skeletonization algorithm Zhang & Suen (1984). Each semantic region is represented as a region node. In contrast, locations connecting multiple semantic regions are represented as junction nodes, thereby forming a region-junction graph that encompasses the entire scene. The edge weights in the graph represent the shortest traversable paths between nodes, computed using the A* algorithm Hart et al. (1968). This graph is online updated throughout the exploration process as new perceptual information is acquired.

Since semantic regions serve as the decision-making units, the system stores a set of RGB frames captured when each region node is recognized, which serve as representative perspectives for subsequent region potential evaluation. On this basis, we design a Scoring-Screening Mechanism (SSM) that integrates the representative perspectives of all semantic regions and information about feasible paths from the agent to each region to assess regional potential. This mechanism then selects the highest-potential target region, thereby guiding the agent's exploration.

Furthermore, to enable dynamic policy adaptation, we further introduce a Dynamic Policy Trigger module. This module employs four triggers to continuously monitor several factors, including changes in the representative perspectives of semantic regions, the current position of the agent in the region-junction graph, and whether new regions are recognized, among others. It dynamically activates the SSM to achieve adaptive adjustment of the exploration policy, as illustrated in Figure 1(**b**) and Figure 6.

Our contributions are presented as follows: (1)We propose DP-Nav, a dynamic navigation framework driven by semantic region potential. This framework represents the entire scene as a region-junction graph for path planning. (2)We introduce the Scoring-Screening Mechanism (SSM), which evaluates and filters each semantic region based on its representative perspectives and the traversability. Subsequently, it prunes the region-junction graph accordingly and allocates the semantic region to navigate. (3) We designed a Dynamic Policy Trigger (DPT) module that employs four triggers to continuously monitor the navigation status, thereby dynamically activating the SSM to enable on-demand policy adaptation. (4) Experiments on ObjectNav, TextNav, and InstanceNav tasks across Gibson, HM3D, and MP3D datasets demonstrate that DP-Nav achieves the SOTA performance.

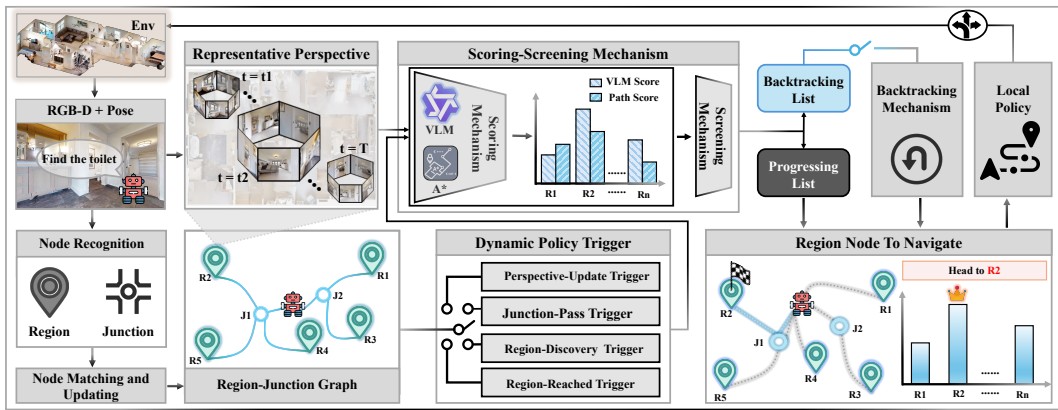

Figure 2: The pipeline of proposed DP-Nav. The detailed workflow is provided in the Overview of the Method section. Notably, when SSM is activated during navigation, only region nodes with representative perspectives updated since the last activation undergo list reallocation.

## 2 RELATED WORK

### 2.1 SCENE REPRESENTATION AND NAVIGATION POLICY FOR VISUAL NAVIGATION

The navigation policy depends on the scene representation. Recent work primarily employs maps Chaplot et al. (2020a); Zhang et al. (2025) or graphs Gu et al. (2024); Yin et al. (2025) for scene representation. On this basis, most zero-shot navigation policies are based on the frontier-based exploration (FBE)Yamauchi (1997). Map-based methods like ESC Zhou et al. (2023) and L3MVN Yu et al. (2023b) utilize semantic maps to select frontiers, while methods like VLFM Yokoyama et al. (2024), OpenFMNavKuang et al. (2024) and InstructNav Long et al. (2024) employ different value maps to represent the semantic association between different regions in the map and the target object, thus guiding the selection of frontiers. Graph-based approaches mainly employ topological maps Krantz et al. (2020); Chaplot et al. (2020b); Zhang et al. (2021); Wu et al. (2024) or construct scene graphs like ConceptGraphs Gu et al. (2024), SG-Nav Yin et al. (2024), and UniGoal Yin et al. (2025) for waypoint or frontier selection. Recent work, such as TriHelper Zhang et al. (2024), also demonstrates "dynamic" capabilities, but its adaptability is manifested through the coordination of three distinct functions: Collision, Exploration, and Detection. In contrast, our approach focuses on policy self-adaptation in response to perceptual feedback and environmental changes.

### 2.2 FOUNDATION MODELS FOR VISUAL NAVIGATION

Current zero-shot navigation frameworks predominantly integrate Large Foundation Models (LFM)Li et al. (2022); Achiam et al. (2023); Bai et al. (2025) to support policy-making. These approaches, such as L3MVN Yu et al. (2023b), ESC Zhou et al. (2023), Co-NavGPT Yu et al. (2023a), and MCoCoNav Shen et al. (2025), convert environmental scene representations into textual descriptions and then leverage LFM for frontier selection. Others like VLFM Yokoyama et al. (2024), OpenFMNav Kuang et al. (2024), and InstructNav Long et al. (2024) utilize LFMs to build semantic relevance between frontiers and the goal based on the agent's egocentric view. Additional methods exemplified by PIVOT Nasiriany et al. (2024), SayPlan Rana et al. (2023), VLMnav Goetting et al. (2024), WMNav Nie et al. (2025), and AO-Planner Chen et al. (2025) adopt end-to-end paradigms to generate waypoints or navigation paths directly from sequential visual observation.

## 3 METHOD

### 3.1 VISUAL NAVIGATION

In visual navigation, the agent is randomly initialized in an unseen environment, perceiving surroundings through an RGB-D sensor while navigating autonomously toward the specified goal.

For ObjectNav, the agent receives an object category and must locate any instance of that category. For TextNav and InstanceNav, the agent receives a description or a reference image of a specific object instance, respectively, and must locate that particular instance. The agent operates with four discrete actions: move_forward, turn_left, turn_right, and stop. Navigation success requires stopping within $w$ meters of the corresponding instance with limited $m$ timesteps.

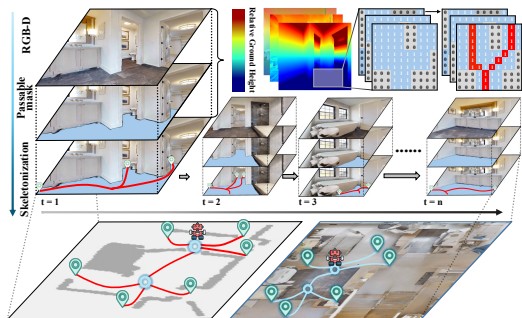

Figure 3: The example of Region-Junction Graph constructing with time. For each timestep, the depth image undergoes passable mask extraction, preprocessing (Detailed in the Appendix A.2), and skeletonization to extract region and junction nodes. In the top-right matrix, values of 1 denote traversable pixels while 0 indicates obstacle.

### 3.2 OVERVIEW

Our DP-Nav pipeline is shown in Figure 2. During navigation, the system recognizes semantic region nodes and junction nodes based on the depth image at each step and updates the graph online. Each semantic region node maintains a representative perspective set that continuously updates during navigation to support policy-making with visual semantic understanding. During policy-making, the Scoring-Screening Mechanism (SSM) evaluates and filters region nodes based on their representative perspective sets and the traversability to the agent. Then, the region nodes are allocated to the Progressing List (PL) or Backtracking List (BL) by SSM. The Progressing (PL nodes exploration) and Backtracking (BL nodes exploration) alternate based on whether the corresponding list is empty. The Progressing takes precedence over Backtracking when both contain unexplored nodes. To enable dynamic policy adjustment, the triggers of the Dynamic Policy Trigger (DPT) module asynchronously activate the SSM, continuously updating both PL and BL. Finally, the local policy Sethian (1996) is responsible for low-level action to the allocated region.

### 3.3 REGION-JUNCTION GRAPH CONSTRUCTION

The scene is represented as a region-junction graph $\mathcal{G} = (\mathcal{V}, \mathcal{E})$ for semantic region-aware dynamic exploration, where $\mathcal{V}$ denotes the node set comprising region nodes $\mathcal{R}$ and junction nodes $\mathcal{J}$. The edge set $\mathcal{E} \subseteq (\mathcal{R} \times \mathcal{J}) \cup (\mathcal{J} \times \mathcal{J})$ captures shortest navigational pathways between these nodes. Next, we elaborate on the region and junction node recognition method with a single-frame depth image. Subsequently, introduce how to update the graph online.

#### 3.3.1 GRAPH NODE RECOGNITION

At each time $t$, given the egocentric depth image $D_t(x, y)$ of resolution $W \times H$, the traversable mask $M_t$ is computed through sensor height $h_{\text{sen}}$, threshold $\tau$, and focal length $f$ derived from horizontal field-of-view $\theta_{\text{hfov}}$ and image width $W$:

$$M_t(x, y) = \mathbb{I}\left[-(y - H/2) \cdot D_t(x, y)/f + h_{\text{sen}} < \tau\right] \tag{1}$$

$$f = W/(2\tan(\theta_{\text{hfov}}/2)) \tag{2}$$

Here, $M_t(x, y)$ is the traversable mask where 1 indicates a navigable pixel, and 0 indicates an obstacle pixel. The term $-(y - H/2) \cdot D_t(x, y)/f$ computes the relative height of each pixel relative to the camera's optical center, offset by the sensor height $h_{\text{sen}}$ above ground level. Please see Figure 3 for visualization. Due to depth noise compromises traversable mask $M_t$ accuracy, necessitating denoising and refinement (Detailed in the Appendix A.2) for the original traversable mask $M_t$ to $M_{cc}$. Then the thinning algorithm $\xi(\cdot)$ Zhang & Suen (1984) extracts a topologically equivalent skeleton, after which redundant short branches are pruned to reinforce the core skeletal structure.

$$S_{\text{pruned}} = \xi(M_{cc}) \setminus \mathcal{D}_{\text{short}} \tag{3}$$

where $\mathcal{D}_{short}$ represents the pruned burr branches (Pruning Detailed in Appendix A.3). Subsequently, Nodes are identified on $S_{\text{pruned}}$ via connectivity patterns:

$$\mathcal{J} = \{p \in S_{\text{pruned}} \mid \deg(p) \geq 3\} \tag{4}$$

$$\mathcal{R} = \{p \in S_{\text{pruned}} \mid \deg(p) = 1\} \tag{5}$$

**Algorithm 1** Node and Graph Updating

**Require:** Nodes $N = \{n_1, n_2, ..., n_k\}$, Graph $G_{t-1}$, Radii $R_r$ and $R_j$, Occupancy map $M_{tocc}$

**Ensure:** Updated graph $G_t$

1: $G_t \leftarrow G_{t-1}$
2: **for** each recognized node $n \in N$ **do**
3:     Transform $n$ to global coordinates
4:     $M \leftarrow \emptyset, r \leftarrow 0$
5:     **if** $n$ is region node **then**
6:        $R_c \leftarrow R_r$
7:        **for** each region node $r \in G_{t-1}$ **do**
8:           **if** $\|r - n\| \leq R_c$ **then**
9:              $M \leftarrow M \cup \{r\}$
10:          **end if**
11:        **end for**
12:     **else**
13:        $R_c \leftarrow R_j$
14:        **for** each junction node $j \in G_{t-1}$ **do**
15:           **if** $\|j - n\| \leq R_c$ **then**
16:              $M \leftarrow M \cup \{j\}$
17:          **end if**
18:        **end for**
19:     **end if**
20:     **if** $M \neq \emptyset$ **then**
21:        $S \leftarrow M \cup \{n\}$
22:        $T \leftarrow \text{traversable}(M_{tocc})$
23:        $P_{fus} \leftarrow \arg\min_{p \in T} \max_{s \in S} \|p - s\|$
24:        $G_{t-1} \leftarrow G_{t-1} \setminus M$
25:        $G_{t-1} \leftarrow G_{t-1} \cup \{P_{fus}\}$
26:        Update edges: $M \rightarrow P_{fus}$
27:     **else**
28:        $G_{t-1} \leftarrow G_{t-1} \cup \{n\}$
29:     **end if**
30: **end for**
31: **return** $G_t \leftarrow G_{t-1}$

where $\deg(p) = \sum_{q \in \mathcal{N}_8(p)} \mathbf{1}_{S_{\text{pruned}}}(q)$ represents the number of skeleton neighbors in its 8-connected neighborhood. Junction nodes $\mathcal{J}$ are defined at points with $\deg(p) \geq 3$, indicating intersections of three or more navigable paths. Region nodes $\mathcal{R}$ correspond to $\deg(p) = 1$ locations, which represent potential access points to adjacent semantic regions.

At each timestep $t$, recognized nodes $N$ are transformed to global coordinates Chaplot et al. (2020a). Then, differentiated matching is performed based on the type of the newly recognized node $N$: if $n \in N$ is a junction node, the existence of any pre-existing junction node within a radius $R_j$ from junction node $n$ triggers node fusion by identifying them as the same physical junction; if $n \in N$ is a region node, the presence of any pre-existing region node within a radius $R_r$ from region node $n$ triggers node fusion by classifying them as a homogeneous functional region. If no matching node exists within the specified search radius, node $n \in N$ is updated into the graph as a new node corresponding to its originally recognized type - either a region node or junction node. For the details above, please refer to Algorithm 1. Then the edge weights $w(n_i, n_j)$ between two nodes are updated as follows:

$$w(n_i, n_j) = \mathcal{L}_{\text{A}^*}(P_{n_i}, P_{n_j} \mid M_{tocc}) \quad (6)$$

Where $w(n_i, n_j)$ denotes the shortest traversable path calculated by the A* algorithm Hart et al. (1968) between $n_i \in \mathcal{V}$ and $n_j \in \mathcal{V}$ ($i \neq j$) based on the occupancy map $M_{tocc}$.

## 3.4 DYNAMIC EXPLORATION DRIVEN BY SEMANTIC REGION POTENTIAL

### 3.4.1 REPRESENTATIVE PERSPECTIVE

During navigation, the same region node $r$ can be recognized from various viewpoints. To enhance region node evaluation for subsequent dynamic policy, an online updating representative perspectives set $\mathcal{V}_r$ is maintained for each region node $r$. Specifically, upon recognizing $r$ at timestep $t$, the system records three key elements from the current perspective: (1) traversable area $A_t$ of node $r$ in current view, (2) corresponding RGB view $I_t$, (3) agent's current pose $Q_t = (P_t, \theta_t)(position, orientation)$. Through continuous screening, the system retains the top $K$ perspectives satisfying conditions $C_1$ and $C_2$, while maximizing the total traversable area. Specifically $C_1$ denotes $\|P_i - P_j\| > \delta_{\text{pos}}$, $C_2$ denotes $\min(|\theta_i - \theta_j|, 360° - |\theta_i - \theta_j|) > \delta_{\text{ang}}$. $C_1$ and $C_2$ drive the agent to sample from multiple locations and angles, enhancing the comprehensiveness of representative perspectives. The updating is formalized as:

$$\mathcal{V}_r = \{(A_i, I_i, P_i) \mid r \text{ recognized}\} \quad (7)$$

$$\mathcal{V}_r^* = \arg\max_{\mathcal{S} \subseteq \mathcal{V}_r, \, |\mathcal{S}| \leq k, \, C_1 \vee C_2} \sum_{s \in \mathcal{S}} A_s \quad (8)$$

Here, $\mathcal{S}$ is a candidate perspectives subset, and $A_s$ is the traversable area of perspective $s$. This generates a compact visual summary $\mathcal{V}_r^*$ of region node $r$ for the subsequent semantic region evaluation.

### 3.4.2 PROGRESSING AND BACKTRACKING

At the task beginning, two lists are initialized: Progressing List $\mathcal{P}$ (PL) and Backtracking List $\mathcal{B}$ (BL) to cache the screened results of region nodes by the following Scoring-Screening Mechanism(SSM). We define PL-node exploration as **Progressing** and BL-node exploration as **Backtracking**. PL-nodes are prioritized for exploration. When PL is empty, Backtracking; when BL is empty, Progressing. Alternate the two until goal discovery. The episode terminates if both are empty. SSM dynamically reallocates PL / BL for environmental changes and perception feedback.

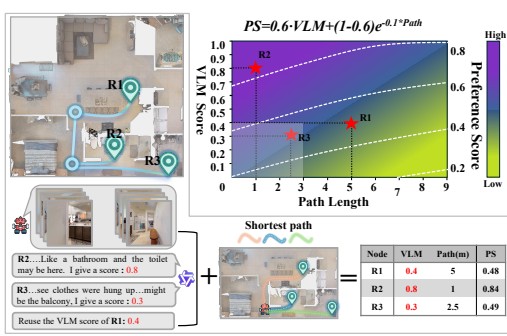

Figure 4: The example of the Scoring-Screening Mechanism evaluates region nodes. The top right subfigure visualizes Eq.11, with light gray regions indicating Backtracking nodes and remaining areas representing Progressing nodes.

### 3.4.3 SCORING-SCREENING MECHANISM

The SSM has two core functions: Scoring and Screening. During the policy-making phase, the system first prompts the VLM to assign a score $\text{VLM}(\mathcal{V}_r^*)$ for each region node $r \in \mathcal{R}$ based on its representative perspectives $\mathcal{V}_r^*$. Subsequently, the shortest navigable distance $D(P_a, P_r) = A^*(P_{\text{agent}}, P_r, M_{\text{tocc}})$ between the agent and each region node is computed via the A* algorithm with the current occupancy map $M_{tocc}$. Then, BL and PL are obtained through Screening. Specifically, region nodes proximal to the agent but with low VLM scores are assigned to BL $\mathcal{B}_t$ via Eq.9. The difference set between all region nodes $\mathcal{R}_t$ and BL then yields PL $\mathcal{P}_t$ (Eq.10).

$$\mathcal{B}_t = \{r \in \mathcal{R} \mid D(P_a, P_r) < \phi \wedge \text{VLM}(\mathcal{V}_r^*) < \zeta\} \tag{9}$$

$$\mathcal{P}_t = \mathcal{R}_t \setminus \mathcal{B}_t \tag{10}$$

Subsequently, the Scoring works for getting preference scores $\mathcal{PS}_r$ of PL nodes via Eq. 11, selecting the highest-scoring node as the next exploration region.

$$\mathcal{PS}_r = \gamma \cdot \text{VLM}(\mathcal{V}_r^*) + (1 - \gamma)e^{-\lambda D(P_{agent}, P_r)} \tag{11}$$

The preference score of each region node combines the VLM score and path length. Where $\gamma \in [0, 1]$ controls the preference weighting, balancing the VLM score against distance sensitivity. $\lambda$ is the distance decay coefficient. For example of SSM scoring and filtering, please refer to Figure 4.

### 3.4.4 DYNAMIC POLICY TRIGGER MODULE

To address the dynamic policy self-adaptation challenge, we designed four simple yet effective triggers to asynchronously activate the SSM for on-demand adjustment of the region node to navigate, as detailed below.

**Region-Discovery Trigger:** This trigger activation occurs through two distinct phases: (1) Initialization phase: At episode beginning, due to the agent's lack of familiarity with the specific environment, it executes a full-circumference scan to establish preliminary situational awareness and construct the initial region-junction graph. Trigger activation follows immediately post-observation. (2) Dynamic navigation phase: During locomotion, evolving vantage points drive representative perspectives update for region nodes, which induces a dynamic change of relative importance across region nodes. The discovery of new region nodes serves as the activation catalyst, triggering a re-evaluation of the semantic region nodes.

**Perspective-Update Trigger:** During navigation, persistent agent movement induces progressive evolution of representative perspectives for region nodes; the cognitive depth toward each region node undergoes continuous refinement. To quantify this evolving perceptual enrichment, we compute the following metric:

$$\eta_t = \frac{1}{K|R_t|} \sum_{r \in R_t} |\Delta \mathcal{V}_r| \tag{12}$$

where $\eta_t \in [0, 1]$ denotes the normalized update intensity; $R_t$ comprises region nodes with updated representative perspectives since the last policy step, and $|\Delta \mathcal{V}_r|$ measures newly updated perspectives for each region node $r$. Furthermore, we define $\varphi_{pug}$ as policy update gate, when $\eta_t > \varphi_{pug}$,

this trigger is activated. Note that the size of policy update gate $\varphi_{pug}$ is negatively correlated to the occurrence of this trigger. The correlation between magnitude $\varphi_{pug}$ and navigation performance is explicated in subsection 4.3.1.

**Junction-Pass Trigger:** Junctions are critical points connecting different regions. However, our policy does not deliberately target junctions. Instead, when an agent goes by a junction node while navigating towards an allocated region, it pauses movement to observe its left and right. This pause facilitates the concurrent update of the region-junction graph and the representative perspectives of relevant nodes, and subsequently, this trigger is activated for policy refinement.

**Region-Reached Trigger:** When the agent reaches the allocated region, its perception of the current region may be incomplete. At this point, the agent observes the left and the right to: (1) confirm whether the goal is present in the current region; (2) update the region-junction graph and representative perspectives of relevant semantic region nodes; If the goal is seen, navigate to it. Or leverage the updated representative perspectives of the current region to prompt the VLM on whether deeper region exploration is required. If the VLM recommends deeper exploration, set a sub-goal within the region and navigate to it. If the goal remains undetected upon sub-goal arrival, activate SSM. If deeper exploration is deemed unnecessary, immediately activate SSM. This trigger's details are listed in Appendix A.4.

The above dynamic triggers persist throughout the entire navigation episode until the agent executes `stop` command or the maximum timesteps are exhausted. When multiple triggers are met simultaneously, they are prioritized as follows: region-reached, region-discovery, perspective-update, and junction-pass. Note that during SSM working, nodes retaining unchanged representative perspectives directly reuse their VLM scores from the previous SSM to reduce computational overhead.

## 4 EXPERIMENTS

### 4.1 SETUP AND IMPLEMENTATION DETAILS

Based on the Habitat platform Savva et al. (2019), we evaluate DP-Nav on ObjectNav across Gibson Xia et al. (2018), HM3D Ramakrishnan et al. (2021), and MP3D Chang et al. (2017) datasets, with TextNav and InstanceNav validated on HM3D. Navigation performance is measured by Success Rate (SR) and Success weighted by Path Length (SPL): $SR = 1/N \sum \mathbb{I}$, $SPL = 1/N \sum \mathbb{I} \cdot \min(1, \ell^*/\ell)$ where $\mathbb{I}$ is the success indicator, $\ell^*$ is the optimal path length, and $\ell$ is the actual path length. SR measures task completion rate, while SPL considers both success and path efficiency.

Our DP-Nav employs Qwen2.5-VL-3B-Instruct Bai et al. (2025) as VLM. The navigation is limited to 500 steps, with a success radius of 0.1 m. The hyperparameters, the processing for different goal specifications, and VLM prompts are provided in the Appendix A.7 A.8 and A.11, respectively.

### 4.2 EXPERIMENT RESULTS AND ANALYSIS

#### 4.2.1 COMPARISON WITH BASELINES

Experimental results in Table 1 demonstrate that our DP-Nav outperforms previous SOTA zero-shot baselines like ApexNav and UniGoal across all three evaluated tasks. Specifically, for the critical HM3D dataset of the ObjectNav task, our DP-Nav outperforms UniGoal by +8% in SR and +10.5% in SPL. For TextNav on HM3D, our DP-Nav surpasses previous SOTA, UniGoal, by +5.4% in SR and +8.1 in SPL. Meanwhile, in the more challenging InstanceNav compared to GOAT and PSL, our DP-Nav exceeds +28.3, +42.7% in SR, and +14.4%, +19.1% in SPL, respectively. Moreover, DP-Nav even surpasses training-based methods on specific datasets, such as SemExp, PONI, GOAT, and Mod-IIN. The performance improvement of DP-Nav could be attributed to the semantic region-aware exploration and on-demand policy adjustment, which avoids ineffective exploration and cross-region backtracking to optimize the path. Furthermore, it saves timesteps to explore more valuable regions, thereby further enhancing the SR. For average improvement compared to the baselines and qualitative analysis, please refer to Figure 5 and Figure 6, respectively.

### 4.3 ABLATION STUDY

| Method | Zero-shot | ObjectNav | | | | | | TextNav | | InstanceNav | |
|---|---|---|---|---|---|---|---|---|---|---|---|
| | | Gibson | | HM3D | | MP3D | | HM3D | | HM3D | |
| | | SR↑ | SPL↑ | SR↑ | SPL↑ | SR↑ | SPL↑ | SR↑ | SPL↑ | SR↑ | SPL↑ |
| SemExp Chaplot et al. (2020a) | ✗ | 65.7 | 33.9 | – | – | 36.0 | 14.4 | – | – | – | – |
| PONI Ramakrishnan et al. (2022) | ✗ | 73.6 | 41.0 | – | – | 31.8 | 12.1 | – | – | – | – |
| IIN-RL-BaselineKrantz et al. (2022) | ✗ | – | – | – | – | – | – | – | – | 8.3 | 3.5 |
| Mod-IIN Krantz et al. (2023) | ✗ | – | – | – | – | – | – | – | – | 56.1 | 23.3 |
| IEVE Lei et al. (2024) | ✗ | – | – | – | – | – | – | – | – | 70.2 | 25.2 |
| PSL Lei et al. (2024) | ✗ | – | – | 42.4 | 19.2 | – | – | 16.5 | 7.5 | 23.0 | 11.4 |
| GOAT Lei et al. (2024) | ✗ | – | – | 50.6 | 24.1 | – | – | 17.0 | 8.8 | 37.4 | 16.1 |
| L3MVN Yu et al. (2023b) | ✓ | 76.9 | 38.8 | 54.2 | 25.5 | – | – | – | – | – | – |
| VoroNav Wu et al. (2024) | ✓ | – | – | 42.0 | 26.0 | – | – | – | – | – | – |
| OpenFMNav Kuang et al. (2024) | ✓ | – | – | 52.5 | 24.1 | 37.2 | 15.7 | – | – | – | – |
| VLFM Yokoyama et al. (2024) | ✓ | 84.0 | 52.2 | 52.5 | 30.4 | 36.4 | 17.5 | – | – | – | – |
| SG-Nav Yin et al. (2024) | ✓ | – | – | 54.0 | 24.9 | 40.2 | 16.0 | – | – | – | – |
| TriHelper Zhang et al. (2024) | ✓ | 85.2 | 43.1 | 56.5 | 25.3 | – | – | – | – | – | – |
| UniGoal Yin et al. (2025) | ✓ | – | – | 54.5 | 25.1 | 41.0 | 16.4 | 20.2 | 11.4 | 60.2 | 23.7 |
| ApexNav Zhang et al. (2025) | ✓ | – | – | 59.6 | 33.0 | 39.2 | 17.8 | – | – | – | – |
| **DP-Nav (Ours)** | ✓ | **88.6** | **62.3** | **62.5** | **35.6** | **45.8** | **25.3** | **25.6** | **19.5** | **65.7** | **30.5** |

Table 1: Comparison of ObjectNav, TextNav, and InstanceNav across Gibson, HM3D, and MP3D datasets. The results demonstrate that our DP-Nav achieves SOTA performance in both path efficiency(SPL) and success rate(SR) compared with baselines.

| | Policy Update Gate $\varphi_{pug}$ | | | | | | | | | | | |
| Rep. Persp. | 0.10 | | 0.20 | | 0.30 | | 0.40 | | 0.50 | | 0.60 | |
| | SR↑ | SPL↑ | SR↑ | SPL↑ | SR↑ | SPL↑ | SR↑ | SPL↑ | SR↑ | SPL↑ | SR↑ | SPL↑ |
|---|---|---|---|---|---|---|---|---|---|---|---|---|
| 3 | 53.2 | 27.8 | 56.2 | 28.5 | 58.0 | 30.1 | 59.1 | 30.6 | 59.0 | 29.1 | 58.1 | 30.3 |
| 4 | 54.3 | 28.3 | 56.3 | 30.7 | 58.7 | 32.8 | 59.6 | 31.1 | 59.6 | 30.8 | 58.4 | 29.7 |
| 5 | 55.4 | 30.6 | 56.8 | 31.3 | 59.2 | 33.1 | 61.2 | 31.9 | 60.5 | 31.2 | 58.7 | 30.4 |
| 6 | 56.3 | 31.1 | 58.3 | 33.4 | 60.7 | 34.6 | **62.5** | **35.6** | 61.1 | 33.8 | 59.3 | 31.5 |

Table 2: Comprehensive parameters ablation on representative perspectives and the policy update gate $\varphi_{pug}$ based on the ObjectNav task of the HM3D dataset.

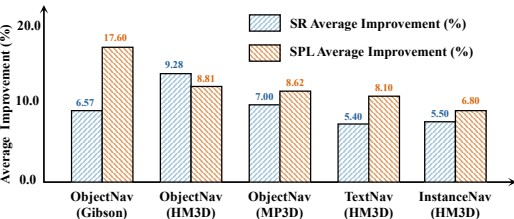

Figure 5: The average improvement in SR and SPL of DP-Nav compared to zero-shot baselines.

Using the ObjectNav task on HM3D as our ablation baseline, we demonstrate the effectiveness of the representative perspective (RP), SSM, and DPT, and investigate the impact of different pipeline ablations on the failure cases.

### 4.3.1 THE IMPACT OF REPRESENTATIVE PERSPECTIVE

To investigate how the representative perspective (RP) quantity and perspective-update trigger frequency affect navigation performance, we configure the maximum RP per region node as {3, 4, 5, 6} while varying the Policy Update Gate $\varphi_{pug}$ of perspective-update trigger (PUT) across {0.1, 0.2, 0.3, 0.4, 0.5, 0.6}. Based on the results in Table 2, we could derive the following preliminary conclusions: (1) The quantity of RP per region node correlates positively with navigation performance, where more viewpoints enhance semantic region assessment robustness while mitigating local observation biases. (2) The $\varphi_{pug}$ requires careful balancing: Excessively low values cause frequent triggering, leading to agent action oscillation and performance drops; whereas excessively high values result in insufficient triggering, inducing agent policy lag and path redundancy.

### 4.3.2 PIPELINE ABLATION AND FAILURE CASE ANALYSIS

We conducted ablation studies on the DPT module and SSM of DP-Nav, with the results presented in Table 3. Notably, since the agent must determine the next region to navigate upon reaching a region, and the Scoring of SSM is necessary for evaluating the semantic region,

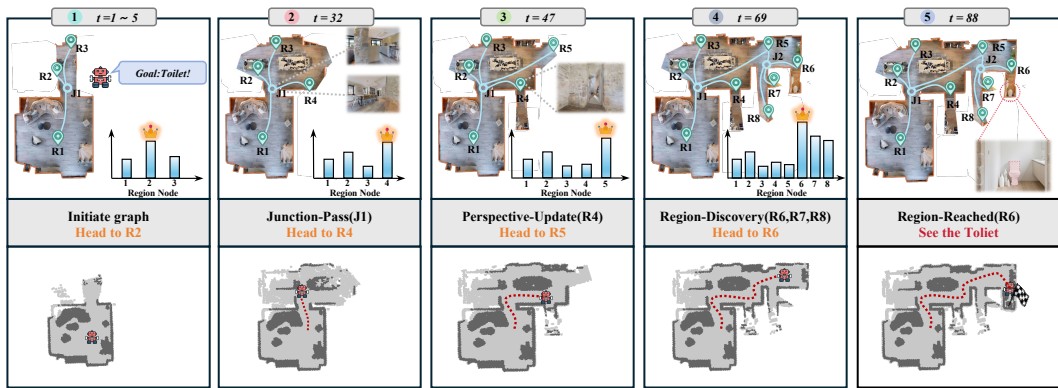

Figure 6: A successful ObjectNav episode by our DP-Nav. The top row depicts the evolving explored environment and corresponding region-junction graph updates throughout navigation. The middle row illustrates policy adaptation with time, while the bottom row displays the agent's trajectory in the occupancy map.

the region-reach trigger and Scoring of SSM are excluded from ablation studies. When the Screening is ablated, the PL/BL distinction is removed, and the region node with the highest score from the Scoring will be explored. Comparing the Triggers ablation with the full pipeline, the SR and SPL typically decrease by -4.6%∼-6.8%, and -3.2%∼-6.5%, respectively. Further, comparing (1) removing both three triggers and Screening, and (2) removing only Screening. We can see that Screening significantly boosts both SR and SPL. Crucially, the ablation of screening alongside the region-discovery trigger causes the most SR and SPL fluctuation(-8.7% SR, -8% SPL). This is because, without Screening, prioritizing nearby low-probability regions wastes timesteps. Moreover, ablating all three triggers and Screening reduces SR by -10.2% and SPL by -7%. Retaining screening alongside ablation improves SR by +1.1% and SPL by +2.2%. The above ablation studies demonstrate that: (1) The Screening enhances SPL and further improves SR by excluding low-relevance proximal regions; (2) Triggers in DPT effectively coordinate with SSM to adapt policy to environmental changes and perception feedback.

**Failure Case Analysis:** To analyze DP-Nav's failure cases and the impact of different components on navigation failures, we defined three types of errors. The experiment results and analysis are presented in Table 4 and Appendix A.9.

| Dynamic Policy Trigger | | | Screening | SR↑ | SPL↑ |
|---|---|---|---|---|---|
| Regi-Dis | Pers-Update | Junc-Pass | | | |
| ✗ | ✓ | ✓ | ✓ | 55.7 | 29.1 |
| ✗ | ✓ | ✓ | ✗ | 53.8 | 27.6 |
| ✓ | ✗ | ✓ | ✓ | 57.0 | 29.0 |
| ✓ | ✗ | ✓ | ✗ | 56.1 | 27.8 |
| ✓ | ✓ | ✗ | ✓ | 57.9 | 32.4 |
| ✓ | ✓ | ✗ | ✗ | 56.2 | 30.8 |
| ✗ | ✗ | ✗ | ✓ | 53.4 | 29.8 |
| ✗ | ✗ | ✗ | ✗ | 52.3 | 27.6. |
| ✓ | ✓ | ✓ | ✗ | 59.5 | 32.2 |
| ✓ | ✓ | ✓ | ✓ | **62.5** | **35.6** |

Table 3: The performance of DP-Nav with different ablations. The "Screening" denotes the Screening mechanism of SSM.

| Ablation | Exploration Error (%)↓ | Detection Error (%)↓ | Planning Error (%)↓ | SR (%)↑ |
|---|---|---|---|---|
| w/o DPT | 14.3 | 24.6 | 7.7 | 53.4 |
| w/o Regi-Dis | 12.3 | 22.3 | 9.7 | 55.7 |
| w/o Pers-Update | 11.4 | 21.6 | 10 | 57.0 |
| w/o Junc-Pass | 10.8 | 22.8 | 8.5 | 57.9 |
| w/o Screening | 10.2 | 23.4 | 6.9 | 59.5 |
| DP-Nav | **6.5** | **20.4** | **10.6** | **62.5** |

Table 4: The failure case quantitative analysis with different ablations. Detection error: missing a visible goal or falsely detecting an absent one. Planning error: failing the task after correct detection, or being stuck within $1m$ for $\geq 400$ steps without goal detection. Exploration error: never locating the goal, without being stuck or false detection. The exploration error rate measures navigation ability toward the goal.

## 5 CONCLUSION AND FUTURE WORK

In this paper, we introduce DP-Nav, a dynamic navigation framework driven by semantic region potential, to improve the path optimization limitations of previous methods' static policies and the problem of cross-region backtracking. Future research will aim for a more flexible navigation policy, more precise semantic region recognition, and more efficient path planning for visual navigation.

# 6 ETHICS STATEMENT

## 6.1 INFORMED CONSENT

Informed consent was obtained from all individual participants involved in this study. The consent process ensured that participants were fully aware of the research purposes, procedures, potential risks, and benefits.

## 6.2 DATA ANONYMITY AND PRIVACY

We implemented strict measures to protect the privacy and anonymity of all participants. All personally identifiable information has been removed or anonymized during data processing and analysis. The data presented in this paper have been aggregated to prevent the identification of any individual participant.

## 6.3 DATA SOURCE AND USAGE

The data utilized in this study were public. The use of this data for research purposes is compliant with the terms of use specified by the data provider and relevant data protection regulations. No unauthorized data collection or usage occurred during this research.

## 6.4 CONFLICT OF INTEREST

The authors of this paper have no financial or non-financial conflicts of interest that might be construed to influence the results or interpretation of the research reported. No funding organization has influenced the design, conduct, analysis, or presentation of this study.

## 6.5 AUTHORSHIP AND ORIGINALITY

We confirm that this manuscript is the original work of the authors and has not been published elsewhere nor is it currently under consideration for publication in any other venue. All authors have contributed significantly to the work and have approved the final version for submission.

## 6.6 BROADER IMPACT STATEMENT

This research aims to advance the field of Embodied Navigation. We have considered the potential societal impacts of our work. While we believe the primary outcomes are beneficial, we acknowledge the importance of ongoing monitoring and discussion regarding the ethical deployment of such technologies to mitigate any potential misuse.

# 7 REPRODUCIBILITY STATEMENT

## 7.1 ALGORITHMIC DETAILS

We provide a comprehensive description of the proposed algorithm, including its core mechanics and theoretical foundations. The paper contains a conceptual outline and pseudocode for the main algorithm to facilitate understanding and re-implementation by other researchers. Key design choices and their justifications are discussed in the manuscript. For a complete, executable implementation of the algorithm and all experiments, please refer to our code in the supplementary materials.

## 7.2 CODE AVAILABILITY

The source code necessary to reproduce all experimental results, including data preprocessing, model training, and evaluation scripts, will be public.

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

## A  APPENDIX

### A.1  USE OF LLMS

LLMs were employed solely as writing aids to polish the language and improve the clarity of expression. They were not used for generating research ideas, designing methods, conducting experiments, or analyzing results. All scientific contributions and substantive content of this work are the sole responsibility of the authors. This use has been disclosed in accordance with the ICLR 2026 policy on LLM usage.

### A.2  TRAVERSABLE MASK PREPROCESSING

Given the initial traversable mask $\mathbf{M}_t \in \{0,1\}^{H \times W}$ from Eq.1, the obtained traversable mask requires preprocessing before skeletonization due to the presence of noise in the depth image and the possibility of multiple independent traversable areas in a single frame. The preprocessing pipeline incorporates structural regularization and selective component retention to enhance path topology while suppressing noise artifacts.

To consolidate fragmented regions and fill small structural gaps while preserving boundary topology, morphological closure is applied:

$$\mathbf{M}_c = \varphi(\mathbf{M}_t; \mathbf{K}_e) \tag{13}$$

where $\varphi$ denotes the morphological closing operator, and $\mathbf{K}_e$ represents an elliptical structuring element defined over integer spatial coordinates $(x, y)$ as:

$$\mathbf{K}_e(x, y) = \mathbb{I}\left[\frac{x^2}{a^2} + \frac{y^2}{b^2} \leq 1\right] \quad \text{with} \quad a = b = 3 \tag{14}$$

where $\mathbb{I}[\cdot]$ is the indicator function. Then, Discrete traversable regions are identified through connected component decomposition using 8-connectivity:

$$\mathcal{C} = \{C_1, \ldots, C_n\} = \mathbf{CC}_8(\mathbf{M}_c) \tag{15}$$

where $\mathbf{CC}_8$ denotes 8-connected component labeling, and each $C_k \subseteq \{1, \ldots, H\} \times \{1, \ldots, W\}$ represents a distinct region with area $A_k = |C_k|$. Subsequently, Component areas are sorted in descending order:

$$A_{(1)} \geq A_{(2)} \geq \cdots \geq A_{(n)} \tag{16}$$

yielding ordered components $C_{(1)}, \ldots, C_{(n)}$. The filtered mask $\mathbf{M}_{\text{cc}}$ is constructed by:

$$\mathbf{M}_{\text{cc}} = \bigcup_{k \in \mathcal{I}} C_{(k)} \tag{17}$$

where the selection index set $\mathcal{I}$ is defined with area threshold $A_{\min} = 100$:

$$\mathcal{I} = \begin{cases} \emptyset & \text{if } n = 0 \\ \{1\} & \text{if } n = 1 \wedge A_{(1)} \geq A_{\min} \\ \{k \in \{1, 2\} \mid A_{(k)} \geq A_{\min}\} & \text{if } n \geq 2 \end{cases} \tag{18}$$

This strategy preserves: (a) no regions when $n = 0$; (b) the largest valid region when $n = 1$; or (c) the two largest valid regions when $n \geq 2$.

The preprocessed binary mask $\mathbf{M}_{\text{cc}}$ then serves as input to skeletonization, having consolidated primary traversable regions while removing noise artifacts.

### A.3 PRUNING DETAILS AFTER SKELETONIZATION

After obtaining the initial skeleton from the traversable region, we apply a two-stage pruning process to enhance skeleton quality. The first stage (Refer to algorithm 2) focuses on removing redundant edges at junction points to eliminate circular paths that do not contribute to navigation. The second stage (Refer to algorithm 3) extends skeleton endpoints to ensure complete coverage of traversable boundaries. Figure. 8 visualizes the comparison before and after pruning.

### A.4 DETAILS OF REGION-REACHED TRIGGER

Upon reaching a semantic region allocated by the *Scoring-Screening Mechanism (SSM)*, the agent first conducts observation by rotating its view left and right to collect $N = 3$ frames from the region. These images are labeled as $\{I_1, I_2, I_3\}$ in sequential order based on the perspective from left to right. The VLM is then prompted to determine whether deep region exploration is necessary based on these images $\mathcal{I} = \{I_i\}_{i=1}^3$, and is required to provide justification for its decision. If the VLM determines that deep exploration is needed, it returns the index $i^*$ of the image containing the optimal exploration direction among the three annotated images. If the VLM determines that deep exploration is unnecessary, the Scoring-Screening Mechanism is directly activated to proceed to the next stage according to the established pipeline. When the VLM confirms the need for deep exploration and returns the optimal exploration direction corresponding to image index $i^*$, the system extracts the traversable region mask $\mathcal{T}_{i^*}$ corresponding to image $I_{i^*}$ based on the depth image and performs preprocessing operations that remain consistent with the node recognition phase. Then, we employ a uniform sampling strategy to select $M \leq 6$ candidate points $\{p_j\}_{j=1}^M$ within the traversable area of the image, with each corresponding position marked with a numerical identifier. Subsequently, the annotated image $I_{i^*}^{annotated}$ along with the previous two original images, are input to the VLM with the designed prompts. Then VLM selects the optimal annotated point $p^*$ from the $M$ candidate

---

**Algorithm 2** Redundant Edge Removal at Junctions

---

**Require:** $skeleton$: initial skeleton image
**Ensure:** $clean\_skeleton$: skeleton with redundant edges removed
1:  $junctions \leftarrow \textsc{FindJunctions}(skeleton)$
2:  $clean\_skeleton \leftarrow skeleton.copy()$
3:  **for** each $(j_x, j_y) \in junctions$ **do**
4:    $branches \leftarrow []$
5:    $visited \leftarrow \textsc{ZerosLike}(skeleton, dtype = bool)$
6:    **for** $(dx, dy) \in \{(-1,-1), (-1,0), \dots, (1,1)\} \setminus \{(0,0)\}$ **do**
7:      $(n_x, n_y) \leftarrow (j_x + dx, j_y + dy)$
8:      **if** $\textsc{IsValid}(n_x, n_y) \wedge skeleton[n_y, n_x] > 0 \wedge \neg visited[n_y, n_x]$ **then**
9:        $branch \leftarrow []$
10:        $queue \leftarrow \textsc{Deque}([(n_x, n_y)])$
11:        $visited[n_y, n_x] \leftarrow true$
12:        **while** $queue \neq \emptyset$ **do**
13:          $(c_x, c_y) \leftarrow queue.\textsc{PopLeft}()$
14:          $branch.\textsc{Append}((c_x, c_y))$
15:          $neighbors \leftarrow 0$
16:          **for** $(ddx, ddy) \in \{(-1,-1), \dots, (1,1)\} \setminus \{(0,0)\}$ **do**
17:            $(nn_x, nn_y) \leftarrow (c_x + ddx, c_y + ddy)$
18:            **if** $\textsc{IsValid}(nn_x, nn_y) \wedge skeleton[nn_y, nn_x] > 0$ **then**
19:              **if** $\neg visited[nn_y, nn_x]$ **then**
20:                $neighbors \leftarrow neighbors + 1$
21:                $queue.\textsc{Append}((nn_x, nn_y))$
22:                $visited[nn_y, nn_x] \leftarrow true$
23:              **end if**
24:            **end if**
25:          **end for**
26:          **if** $neighbors = 0$ **then**
27:            **break**
28:          **end if**
29:        **end while**
30:        $branches.\textsc{Append}(branch)$
31:      **end if**
32:    **end for**
33:    **if** $|branches| > 2$ **then**
34:      $shortest\_branch \leftarrow \arg\min_{b \in branches} |b|$
35:      **for** $i \leftarrow 1$ **to** $|shortest\_branch| - 1$ **do**
36:        $(x, y) \leftarrow shortest\_branch[i]$
37:        $clean\_skeleton[y, x] \leftarrow 0$
38:      **end for**
39:    **end if**
40: **end for**
41: **return** $clean\_skeleton$

---

points based on the content features exhibited by these images. The selected point is projected onto the global map, and then the local policy executes navigation to it. If the goal remains undetected after reaching the point and scanning the surroundings, SSM activates for subsequent navigation phases.

However, the above-mentioned process requires calling VLM twice, which increases the time and computational costs. In the specific experiment, we combined the two steps into one; that is, before prompting the VLM, we sampled each image and labeled the corresponding points for the VLM to reason. Please see the Figure.7 for an example. *VLM Prompts* subsection A.11.

---

**Algorithm 3** Skeleton Extension from Endpoints

---

**Require:** $skeleton$: cleaned skeleton image
**Require:** $traversable\_mask$: binary traversable region mask
**Require:** $max\_extension$: maximum extension distance
**Ensure:** $extend\_skeleton$: skeleton with extended endpoints
1: $endpoints \leftarrow$ FINDENDPOINTS($skeleton$)
2: $extend\_skeleton \leftarrow skeleton.copy()$
3: **for** each $(x, y) \in endpoints$ **do**
4:     SKNEI = GETSKELETONNEIGHBORS($x, y, skeleton$)
5:     $neighbors \leftarrow$ SKNEI
6:     **if** $|neighbors| = 1$ **then**
7:         $(dx, dy) \leftarrow neighbors[0]$
8:         $(ext\_dx, ext\_dy) \leftarrow (-dx, -dy)$
9:         **for** $i \leftarrow 1$ **to** $max\_extension$ **do**
10:            $(n_x, n_y) \leftarrow (x + ext\_dx \cdot i, y + ext\_dy \cdot i)$
11:            **if** $\neg$ISVALID($n_x, n_y$) $\lor traversable\_mask[n_y, n_x] = 0$ **then**
12:                $end\_point \leftarrow (x + ext\_dx \cdot (i-1), y + ext\_dy \cdot (i-1))$
13:                DRAWLINE($extend\_skeleton, (x, y), end\_point$)
14:                **break**
15:            **end if**
16:         **end for**
17:     **end if**
18: **end for**
19: **return** $extend\_skeleton$

---

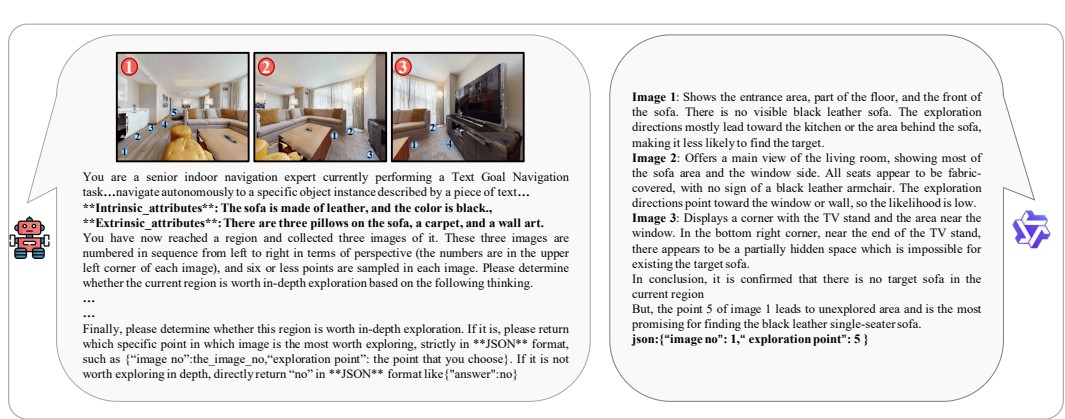

Figure 7: The example(TextNav) of prompting the VLM to determine if deeper exploration is required after the agent reached the region allocated by the Scoring-Screening Mechanism. If the VLM deems that the current region requires in-depth exploration, it will return the pre-annotated number in the corresponding perspective. Then, we map this number to the global occupancy map, and the local policy is responsible for reaching the point. If the VLM deems that the region does not require in-depth exploration, then the next stage will be carried out according to our DP-Nav pipeline; that is, the SSM will allocate the next region to navigate.

## A.5 DATASETS DETAILS

**ObjectNav**: We conduct experiments on three scene datasets: Gibson (5 scenes with 1,000 episodes covering 6 object categories), MP3D (11 scenes with 2,195 episodes covering 21 categories), and HM3D (20 scenes with 2,000 episodes covering 6 categories). **InstanceNav**: The InstanceNav datasets based on HM3D have 795 unique instances in 1000 test episodes. Each episode in the InstanceNav datasets corresponds to a unique goal object instance. **TextNav**: The TextNav dataset extends the InstanceNav dataset, containing 795 distinct instances across 20 scenes. Each instance is annotated through dual attribute categories: (1) *Intrinsic Attributes* describing inherent object

properties (e.g., shape, color, material); (2) *Extrinsic Attributes* capturing environmental context, enabling precise differentiation between instances sharing similar intrinsic features.

## A.6 BASELINES DETAILS

The Frontier-Based Exploration (FBE) policy, pioneered by Yamauchi (1997), focuses on identifying environmental frontiers — boundaries between known and unknown regions — to guide the agent in incrementally exploring unmapped spaces. Recent research has optimized exploration efficiency through diversified frontier selection algorithms. The following baselines are typical methods based on FBE.

- L3MVN Yu et al. (2023b): L3MVN leverages a semantic map to extract object categories near frontiers. It then employs a large language model to infer semantic relationships between these categories and the target object, ultimately selecting the frontier with the highest semantic relevance for agent exploration.

- OpenFMNav Kuang et al. (2024): OpenFMNav integrates multiple foundation models to generate a Versatile Semantic Score Map (VSSM). This map dynamically encodes object semantics with confidence scores. The system evaluates semantic relevance between frontiers and goal objects through VSSM, enabling optimal frontier selection for exploration.

- VLFM Yokoyama et al. (2024): VLFM constructs a language-grounded value map by computing semantic correlations between sequential RGB observations and the goal object using the BLIP-2 Li et al. (2022) language model. Navigation decisions are made by selecting frontiers exhibiting maximal semantic alignment with the goal within this map.

- SG-Nav Yin et al. (2024) and UniGoal Yin et al. (2025): SG-Nav and UniGoal both employ 3D scene graphs to model environmental object relationships. During navigation, they compute semantic similarity between sub-graphs near frontiers and the goal category, subsequently guiding the agent toward frontiers with the strongest semantic correspondence.

- Trihelper Zhang et al. (2024): Trihelper dynamically integrates three dedicated helpers to address collisions, inefficient exploration, and target misidentification in object goal navigation. Specifically, a Collision Helper that uses clustering to redirect the agent from trapped positions; An Exploration Helper that monitors goal proximity to trigger exploratory behavior when progress stalls; A Detection Helper that leverages vision-language models to verify target objects and reduce false positives.

- ApexNavZhang et al. (2025): ApexNav proposes an adaptive exploration strategy—dynamically leveraging semantic or geometric cues based on environmental semantics—and a target-centric semantic fusion method for accurate object identification under noisy detections.

## A.7 HYPERPARAMETERS

The hyperparameters of our DP-Nav could be found in Table 5. Note that, except for the parameter ablation experiments, all other experiments are based on these hyperparameters.

## A.8 PROCESSING OF DIFFERENT GOAL SPECIFICATION

We conduct experiments on three goal-oriented tasks: Object Goal Navigation (ObjectNav), Text Goal Navigation (TextNav), and Instance Image Goal Navigation (InstanceNav). In ObjectNav, the agent is required to navigate to any instance of a specified object category. For TextNav and InstanceNav, the agent is provided with a textual description of the goal instance and its surrounding layout or an image containing a specific goal instance, respectively, and must navigate to that specific instance.

The goal specification of the above three tasks involves two modalities (text and visual). For the convenience of unified processing, we directly combine the goal text (ObjectNav, TextNav) or image(InstanceNav) with the designed prompt to guide the VLM to score each region node based on the corresponding representative perspectives. The designed prompts could be found in the following

| Parameter | Value |
|---|---|
| camera-relative traversable distance $\tau$ | 0.8749m |
| region node matching radius $R_r$ | 0.65m |
| junction node matching radius $R_r$ | 1.2m |
| The number of RP $K$ | 6 |
| Policy Update Gate $\varphi$ | 0.4 |
| RP update location change $\delta_{\text{pos}}$ | 0.5m |
| RP update orientation difference $\delta_{\text{ang}}$ | 30° |
| region node screening path length $\phi$ | 3m |
| region node screening VLM score $\zeta$ | 0.4 |
| preference score weight $\gamma$ | 0.6 |
| preference score distance decay coefficient $\lambda$ | 0.1 |
| RGB-D sensor height $h_{sen}$ | 0.88m |
| agent radius | 0.18m |
| horizontal field-of-view $\theta_{hfov}$ | 79 |
| allow sliding | false |
| frame height | 480 |
| frame width | 640 |

Table 5: Hyperparameters of our *DP-Nav* for experiments. RP denotes *Representative Perspective* of each region node.

A.9 FAILURE CASE ANALYSIS

To analyze DP-Nav's failure cases and the impact of different components on navigation failures, we defined three types of errors. Detection error: missing a visible goal or falsely detecting an absent one. Planning error: failing the task after correct detection, or being stuck within $1m$ for $\geq 400$ steps without goal detection. Exploration error: never locating the goal, without being stuck or false detection. The exploration error rate measures navigation ability toward the goal. As illustrated in Table 4, the removal of individual triggers in the Dynamic Policy Trigger (DPT) module results in increased exploration errors of +8% and +9%, whereas abolishing the entire DPT module causes the most significant degradation, with exploration error rising to 14.3% (Compared to DP-Nav's 6.5%). This unequivocally validates the effectiveness of our dynamic exploration policy. Moreover, DP-Nav exhibits a detection error $21\% \sim 25\%$, which remains higher than the exploration error. The primary reason lies in DP-Nav's key innovation: a semantic region potential-driven dynamic navigation framework designed to overcome inherent issues in prior approaches, including rigidity in static policies and inefficient backtracking across regions, ultimately improving navigation performance and path efficiency.

A.10 MORE EXPERIMENTAL VISUALIZATION

We visualize three episodes for each task, respectively. Please see the Figure. 9 (ObjectNav), Figure. 10 (TextNav), Figure. 11 (Instancenav). In addition, we have compiled navigation videos for the three goal-oriented navigation tasks, which can be found in the supplementary materials.

A.11 VLM PROMPTS

We mainly call VLM in the following two situations. The first is to prompt the VLM to score the region node based on the corresponding representative perspective and goal specification. This Prompt is called *Score Prompt*. The second is in the *Region-Reached Trigger* of *Dynamic Policy Trigger* module. When the agent reaches a region node assigned by the *Scoring-Screening Mechanism*, based on the corresponding observations of the reached region, we prompt the VLM to deter-

mine whether deeper exploration is required in this region. We call this prompt *Advice Prompt*. The details of *Score Prompt* and *Advise Prompt* could be found in the Figure. 12 (ObjectNav), Figure. 13 (TextNav), and Figure. 14 (InstanceNav).

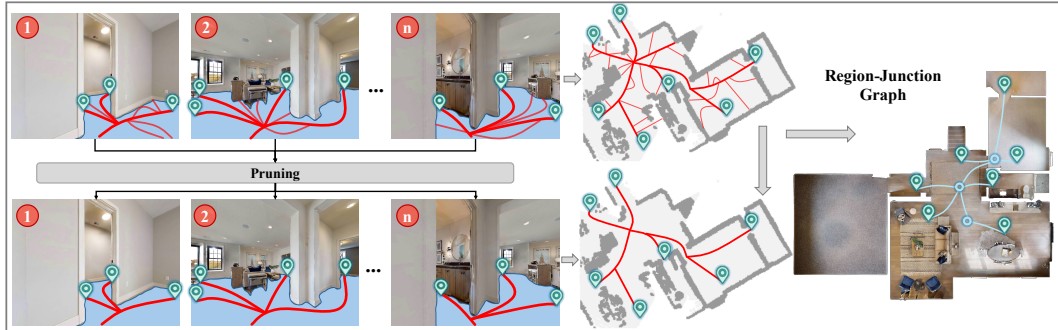

Figure 8: After obtaining the depth image at each time step T= 1,2...n, the traversable area is extracted, preprocessed, skeletonized, and finally pruned. The pruning operations mainly remove excess burrs and shorter branches to highlight the main topological structure. This image shows the comparison before and after pruning, as well as the online updated region-junction graph.

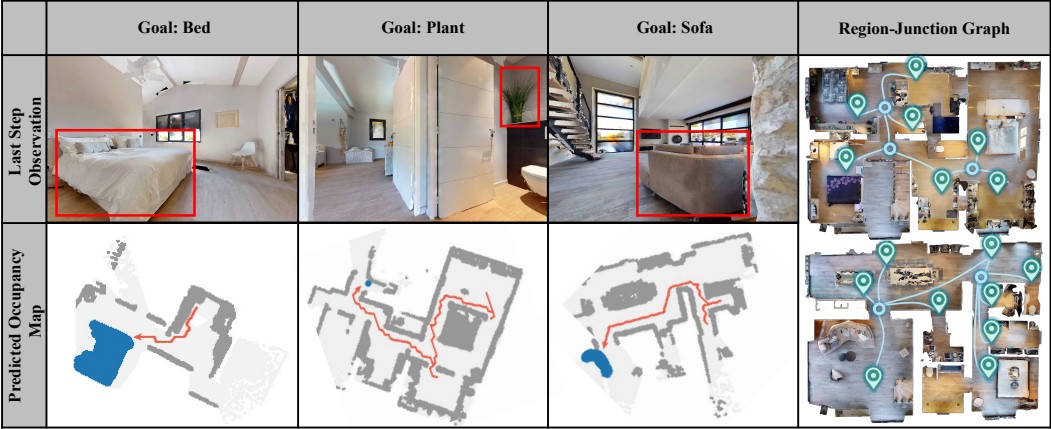

Figure 9: A Successful navigation episode of ObjectNav by our DP-Nav.

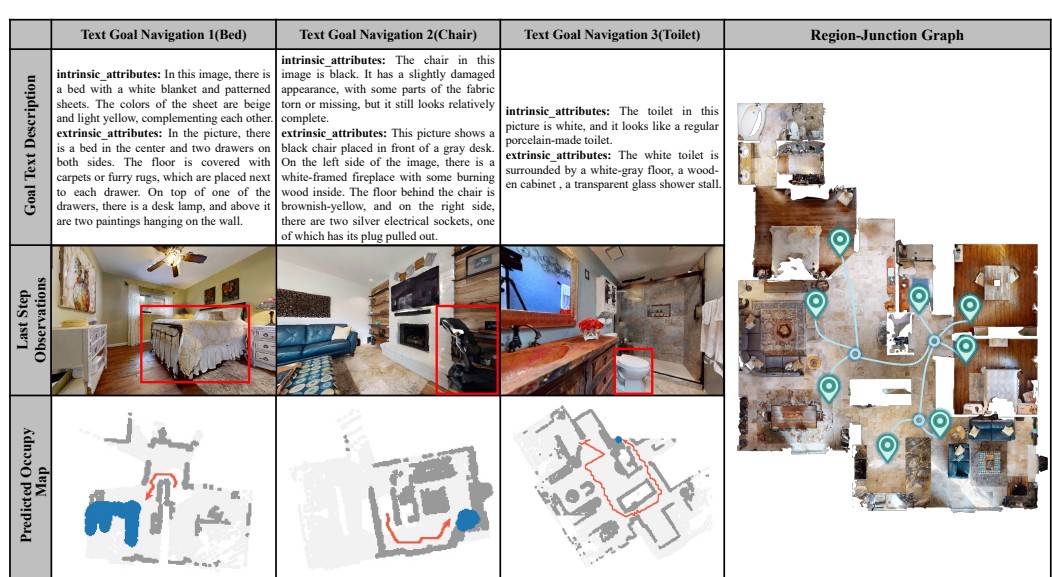

Figure 10: A Successful navigation episode of TextNav by our DP-Nav.

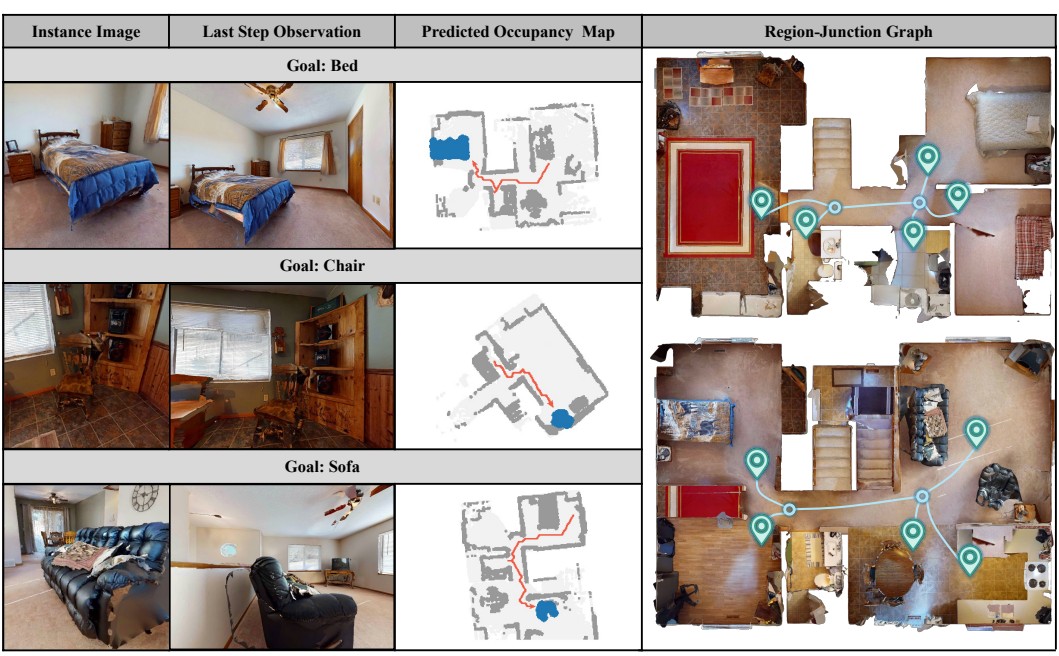

Figure 11: A successful navigation episode of InstanceNav by our DP-Nav.

### Score Prompt

You are a senior indoor navigation expert currently performing an Object Goal Navigation task. Your starting position is randomly initialized in an unfamiliar environment. The goal category **bed** has been specified. You are required to perceive the environment using RGB visual inputs and autonomously navigate to any instance of the **bed** category.

Now RGB observations from a candidate region are provided. Based on the visual information, evaluate the goal presence probability. You should consider but not be limited to :

1. **Don't just focus on whether the goal object appears in the picture, you should consider co-occurrence probability between goal object and other objects**

2. **Deduce region type from visual features to evaluate goal presence likelihood **

3. **Assess exploration worthiness if the region is only partially visible**

4. **Other information and principles that you consider useful**

5. **Assign higher probability when goal object appear in view**

Finally, return a probability score within **[0,1]** (1 indicates highest likelihood) in strict **JSON ** format: {"score": your_float_score}

### Advice Prompt

You are a senior indoor navigation expert currently performing an Object Goal Navigation task. Your starting position is randomly initialized in an unfamiliar environment. The goal category**bed** has been specified. You are required to perceive the environment using RGB visual inputs and autonomously navigate to any instance of the **bed** category.

You have now reached a region and collected three images of it. These three images are numbered in sequence from left to right in terms of perspective (the numbers are in the upper left corner of each image), and six or less points are sampled in each image. Please determine whether the current region is worth in-depth exploration based on the following thinking.

Decision-making should consider but not be limited to:

1. ** Don't just focus on whether the goal object appears in the picture , you also should consider co-occurrence probability between goal objects and other objects**

2. **Deduce region type from visual features to evaluate goal presence likelihood**

3. ** Select candidate exploration point closest to goal when visible **

4. ** If goal not visible but passages exist, evaluate if they may lead to potential goal areas **

5. ** Other beneficial principles or information **

Finally, please determine whether this region is worth in-depth exploration. If it is, please return which specific point in which image is the most worth exploring, strictly in **JSON** format, such as {"image no":the_image_no,"exploration point": the point that you choose}. If it is not worth exploring in depth, directly return "no" in **JSON** format like {"answer":no}

Figure 12: The example prompt of ObjectNav for finding a bed.

**Score Prompt**

You are a senior indoor navigation expert currently performing a Text Goal Navigation task. Your starting position is randomly initialized in an unfamiliar environment. Your task is to perceive the environment through RGB information and navigate autonomously to a specific object instance described by a piece of text. It should be noted that this object instance must exist in this environment. The current task's goal instance is a bed, and the description of this specific bed has the following two aspects: "**1. Intrinsic attributes**: there is a bed with a white blanket and patterned sheets. The colors of the sheet are beige and light yellow, complementing each other; **2. Extrinsic_attributes**: there is a bed in the center and two drawers on both sides. The floor is covered with carpets or furry rugs, which are placed next to each drawer. On top of one of the drawers, there is a desk lamp, and above it are two paintings hanging on the wall."

Now RGB observations from a candidate region are provided. Based on the visual information of the candidate region, evaluate the goal presence probability in the candidate region. You should consider but not be limited to :

1. **You should not only focus on the goal object bed itself, but also pay attention to the environmental layout around the bed as described above, as well as the attributes of the goal object and its surroundings such as color and material**.

2. **Don't just focus on whether the goal object appears in the picture, you should consider co-occurrence probability between goal object and other objects**

3. **Deduce region type from visual features to evaluate goal presence likelihood **

4. **Assess exploration worthiness if the region is only partially visible**

5. **Other information and principles that you consider useful**

Finally, return a probability score within **[0,1]** (1 indicates highest likelihood) in strict **JSON ** format: {"score": your_float_score}

**Advice Prompt**

You are a senior indoor navigation expert currently performing an Text Goal Navigation task. Your starting position is randomly initialized in an unfamiliar environment. Your task is to perceive the environment through RGB information and autonomously navigate to a specific object instance described by a piece of text. It should be noted that this object instance must exist in this environment. The current task's goal category is **bed**, and the description of this specific bed has the following two aspects: "**1. Intrinsic attributes**: there is a bed with a white blanket and patterned sheets. The colors of the sheet are beige and light yellow, complementing each other; **2. Extrinsic_attributes**: there is a bed in the center and two drawers on both sides. The floor is covered with carpets or furry rugs, which are placed next to each drawer. On top of one of the drawers, there is a desk lamp, and above it are two paintings hanging on the wall."

You have now reached a region and collected three images of it. These three images are numbered in sequence from left to right in terms of perspective (the numbers are in the upper left corner of each image), and six or less points are sampled in each image. Please determine whether the current region is worth in-depth exploration based on the following thinking.

Decision-making should consider but not be limited to:

1. You should not only consider the specific object instance, but also take into account the environmental layout around the object described above.

2. **Don't just focus on whether the goal object appears in the picture, you should consider co-occurrence probability between goal object and other objects.**

3. **Deduced region type from visual features to evaluate goal presence likelihood.**

4. ** Select candidate exploration point closest to specific goal instance when you are sure the the specific bed instance appear. **

5. ** If the specific goal instance is not visible but passages exist, evaluate if they may lead to potential goal areas.**

6. ** Other beneficial principles or information.**

Finally, please determine whether this region is worth in-depth exploration. If it is, please return which specific point in which image is the most worth exploring, strictly in **JSON** format, such as {"image no":the_image_no,"exploration point": the point that you choose}. If it is not worth exploring in depth, directly return "no" in **JSON** format like{"answer":no}

Figure 13: The example prompt of TextNav for a specific bed instance.



**Score Prompt**

You are a senior indoor navigation expert currently performing an Insance ImageGoal Navigation task. Your starting position is randomly initialized in an unfamiliar environment. Your task is to perceive the environment through RGB information and navigate autonomously to a specific object instance specified by a goal image. This picture shows a specific instance of an object in this environment. It should be noted that this object instance must exist in this environment. In this task, the object instance is a **bed**, and the **instance goal image** has been given to you.

Now RGB observations from a candidate region are provided. Based on the visual information of the candidate region, evaluate the goal presence probability in the candidate region. You should consider but not be limited to :

1. **You should not only focus on the goal object bed itself, but also pay attention to the environmental layout around the bed as described in the goal image, as well as the attributes of the goal instance and its surroundings such as color and material**.

2. **Don't just focus on whether the goal object appears in the picture, you should also consider co-occurrence probability between goal object and other objects**

3. **Deduce region type from visual features to evaluate goal presence likelihood **

4. **Assess exploration worthiness if the region is only partially visible**

5. **Other information and principles that you consider useful**

Finally, return a probability score within **[0,1]** (1 indicates highest likelihood) in strict **JSON ** format: {"score": your_float_score}

**Advice Prompt**

You are a senior indoor navigation expert currently performing a Insance ImageGoal Navigation task. Your starting position is randomly initialized in an unfamiliar environment. Your task is to perceive the environment through RGB information and navigate autonomously to a specific object instance specified by a goal image. This picture shows a specific instance of an object in this environment. It should be noted that this object instance must exist in this environment. In this task, the object instance is a **bed**, and the **instance goal image** has been given to you.

You have now reached a region and collected three images of it. These three images are numbered in sequence from left to right in terms of perspective (the numbers are in the upper left corner of each image), and six or less points are sampled in each image. Please determine whether the current region is worth in-depth exploration based on the following thinking.

Decision-making should consider but not be limited to:

1.You should not only consider the specific object instance, but also take into account the environmental layout around the object described above.

2. **Don't just focus on whether the goal object appears in the picture, you should consider co-occurrence probability between goal object and other objects.**

3. **Deduced region type from visual features to evaluate goal presence likelihood.**

4. ** Select candidate exploration point closest to specific goal instance when you are sure the the specific bed instance appear. **

5. ** If the specific goal instance is not visible but passages exist, evaluate if they may lead to potential goal areas.**

6. ** Other beneficial principles or information.**

Finally, please determine whether this region is worth in-depth exploration. If it is, please return which specific point in which image is the most worth exploring, strictly in **JSON** format, such as {"image no":the_image_no,"exploration point": the point that you choose}. If it is not worth exploring in depth, directly return "no" in **JSON** format like{"answer":no}



Figure 14: The example prompt of InstanceNav for a specific bed instance.

