# OpenReview forum: "DP-Nav: Dynamic Exploration Driven by Semantic Region Potential for Zero-shot Visual Navigation"
_ICLR.cc/2026/Conference — Submitted to ICLR 2026_

### Official Review · Reviewer_8dHA · 2025-10-29

**Soundness:** 2
**Presentation:** 3
**Contribution:** 2
**Rating:** 4
**Confidence:** 5

**Summary:**

The paper proposes DP-Nav, a dynamic visual navigation framework driven by semantic region potential. It represents the scene with a “region–junction graph” and aggregates each region via representative views. At the policy level, a Scoring–Screening Mechanism (SSM) fuses VLM semantic scores with path cost, and four Dynamic Policy Triggers (DPT) enable on-demand replanning. Experiments on ObjectNav, TextNav, and InstanceNav in Gibson/HM3D/MP3D report improvements over zero-shot baselines along with ablations.

**Strengths:**

* **Cross-task and cross-dataset zero-shot comparisons**
  • On HM3D ObjectNav, DP-Nav outperforms UniGoal by +8% SR and +10.5% SPL.
  • On HM3D TextNav, it surpasses UniGoal by +5.4% SR and +8.1% SPL.
  • On InstanceNav, it shows substantial gains over GOAT and PSL.

* **Relatively complete implementation and appendix details**
  • The paper specifies the VLM used (Qwen2.5-VL-3B-Instruct) and the step limit / success radius.
  • The appendix explains prompt usage scenarios (scoring prompts and deep-exploration prompts).

**Weaknesses:**

* **Insufficient direct validation that VLM scoring reflects “navigability”**
  • The textual example in Fig. 4 (“Looks like a restroom, give 0.8”) appears to measure semantic visibility rather than reachability.
  • SSM treats $VLM(V*_r)$ as the core of region potential, yet provides no correlation or calibration curves versus success rate or SPL.
  • Only one VLM model is used; there is no cross-model consistency or degraded controls (e.g., random or heuristic scores).

* **Coverage and omissions of hand-crafted triggers are not systematically evaluated**
  • The four triggers are rule-based; despite gains, there is no quantitative diagnostic for misses and false alarms.
  • The view-update gate $φ_pug$ being too small induces “action oscillation,” while too large causes “policy lag,” indicating strong hyperparameter sensitivity.
  • No specific analysis is provided for difficult layouts such as long corridors, loops, dead ends, or repeated visits.
  • For the Region-Reached trigger that re-invokes the VLM to decide “deep exploration,” the false-trigger rate and backtracking cost are unreported.

* **Limited interpretability and sensitivity analysis for SSM weighting and screening**
  • For preference score ( $\mathrm{PS}=\gamma\cdot \mathrm{VLM}+(1-\gamma)e^{-\lambda\cdot \mathrm{Path}}$ ), γ and λ appear only in a table without theoretical or empirical calibration procedure.
  • The assignment to progressing/backtracking lists depends on threshold sets (ϕ, ζ), but systematic sensitivity curves are missing.

* **Experimental fairness and statistical rigor need strengthening**
  • The SOTA claims in Table 1 lack variance, confidence intervals, and multiple random seeds.
  • Although there are ablations on triggers and SSM, significance testing and stratified comparisons across scenarios are absent.

**Questions:**

* **Validate and calibrate VLM scoring for navigability**
  • Construct counterfactual scenes that are semantically visible yet hard to reach, and report correlations and segmented trends between VLM scores and success rate/SPL. Provide an ROC curve for ζ with target AUC above random.
  • Conduct cross-model consistency experiments (add 1–2 more VLMs) and degraded controls (random scores, simple heuristics) to verify robust ranking. Target an improvement of at least x% over degraded controls, presented as new columns in a Table 1–style summary.
  • Report sensitivity scans of ζ and performance under OOD conditions (occlusion, camouflage, atypical appearances); provide threshold–performance curves and select a stable operating point.
  • In Appendix A.11, include full prompts, few-shot examples, failure cases, and statistics of failure types (false positives/negatives) aligned with the subjective text in Fig. 4.

* **Systematically evaluate trigger coverage to reduce omissions and false triggers**
  • Build benchmark sets for long corridors, loops, dead ends, and repeated visits; report trigger rates, false-trigger rates, and miss rates for each of the four triggers with PR curves or box plots.
  • Perform fine-grained scans over $φ_pug$ and couple them with different numbers of representative views (RP) to quantify “oscillation” versus “lag,” reporting policy-switch frequency and average path redundancy.
  • Add a “trigger failure type” column to a Table 3/4–style ablation, quantifying each failure category and its contribution to SR/SPL to verify improved coverage.

* **Enhance experimental fairness and statistical rigor**
  • For all entries in Table 1, report mean ± standard deviation with 3–5 random seeds.

* **Assess VLM output consistency and sampling variance (same environment, different sampling)**
  • Fix the environment and camera pose; vary the VLM random seed, temperature, or sampling strategy (top-k, top-p). Re-score multiple times and report consistency of region scores and rankings using ICC, Kendall τ, or Spearman ρ. Criterion: τ or ρ ≥ 0.8; if lower, analyze causes.
  • Apply small pose jitter within the same environment (translation ±5 cm, rotation ±5°), resample representative views, and replicate scoring. Compute per-region score variance and coefficient of variation; plot variance versus jitter magnitude. Criterion: within task tolerance, score variance should not cause frequent flips among the top-k regions.
  • Test prompt robustness via minor paraphrases and synonym substitutions; compare ranking differences and report the similarity distribution. Criterion: high median similarity with a small interquartile range.
  • Quantify the effect of consistency on navigation outcomes by comparing the distributions of SR and SPL across repeated scoring-driven plans. Report mean differences and confidence intervals and the maximum observed fluctuation. Criterion: SR/SPL variation remains within a preset bound, for example absolute difference ≤ 1 percentage point.

---

### Official Review · Reviewer_ZTxk · 2025-10-30

**Soundness:** 3
**Presentation:** 2
**Contribution:** 2
**Rating:** 2
**Confidence:** 3

**Summary:**

This paper proposes a method for constructing a topological graph for object-goal navigation. This graph consists of two types of nodes: region nodes and junction nodes, where region nodes represent areas to explore and junction nodes connect region nodes. A Score-Screening Mechanism (SSM) is proposed using LVM to assign scores to each region node and divide them into a processing list and a backtracking list. A Dynamic Policy Trigger (DPT) module is introduced to activate SSM and dynamically drive the agent's exploration. Experiments are conducted on three types of navigation tasks across three datasets, achieving superior performance.

**Strengths:**

1. This paper identifies two issues in existing frontier-based exploration (FEB) navigation: frontiers are updated at fixed time steps, and the agent may jump inconsistently between different regions due to each frontier being treated individually.

2. The method part is technically sound. Note that it refers to the correctness and feasibility, with a clear illustration of nice-looking figures.

3. The experiment performance is superior to the recent works on all three tasks across different datasets. In some cases, the SR and SPL improvement is evident by a margin.

**Weaknesses:**

There are several weaknesses concerning novelty and claim:

1. Sections 3.3.1 to 3.4.3 are more engineering-oriented; that is, it is hard to identify new network architectures, algorithms, or pipelines. This is not questionable with respect to the correctness of the method, but rather concerns the novelty. The authors should highlight the key novelty that is different from previous works.

2. In Related Work (Line 141 - 142), the authors briefly mention that the proposed method is a policy with self-adaptation, which is different from previous graph-based methods. However, it lacks support or evidence for "self-adaptation" in the Method section. The main distinction from previous graph-based methods is not highlighted.

3. The authors mention two key issues in existing FBE methods in the Introduction (Line 49 - 76), but how the proposed method addresses these two issues is not clearly stated. For example, does the DPT module update frontiers at varying time steps? Do we only consider one frontier in each semantic region, or consider all frontiers in the semantic region as a whole for the decision?

4. The authors mention "semantic" regions several times. But in Section 3.3.1, the region node is constructed purely based on geometry with traditional approaches. It is questionable these regions truly contain meaningful semantics.

5. The four triggers described in Section 3.4.4 are not clearly presented. It is hard to follow this section. I regard this as a writing issue. In addition, it is unclear whether others can borrow this idea without using the graph built in Section 3.3.1, as these triggers closely coupled with the graph.

**Questions:**

1. Can we merge two junction nodes in the topological graph regardless of their distance? In Line 229 - 230, there is a radius threshold to fuse junction nodes, which I think can be removed.

2. Why do we need to differentiate the Processing list and the Backtracking list? We can just explore regions based on the combination of VLM scores and distance in descending order.

---

> ### Author Response · Authors · 2025-11-21
> **Response for Weakness 1,2,3**
>
> ## Weakness 1
> First, we ***acknowledge*** the core of your point—that the contributions in Sections 3.3.1 to 3.4.3 are ***not a novel network architecture.*** This is a key point for our work. In the field of visual navigation, particularly within the ***Zero-Shot setting*** that leverages powerful pre-trained foundation models (like VLMs), we observed that the performance bottleneck often ***resides not in the capabilities of the models themselves***, but in ***how to effectively integrate these capabilities into the navigation pipeline*** of an embodied agent.
>
> Thus, the design philosophy behind DP-Nav is: ***Rather than designing a new network from scratch***, we focus on constructing ***a novel decision-making pipeline that maximizes the potential of existing foundation models.*** We believe the ***"novelty" ***of a method can be demonstrated at*** multiple levels***, including:
>
> - ***Semantic Region as Exploration Unit:*** We are the **first** to propose treating ***"semantic regions"*** as the ***fundamental unit for exploration,*** instead of the ***traditional "frontiers.***" This fundamentally alters the granularity of exploration, aiming to address the long-standing challenges of ***cross-region backtracking and path redundancy.***
>
> - ***Dynamic Policy:*** We designed the unique Scoring-Screening Mechanism (SSM) and Dynamic Policy Trigger (DPT) modules. It enables the agent to ***adjust its strategy*** in real-time based on ***environmental feedback,*** much like a human would, ***overcoming*** the limitation of ***static policies prevalent in existing methods.*** The innovativeness of this ***dynamic interaction*** process is a significant ***contribution*** of the paper.
>
> ***In conclusion***, we firmly believe that in this era of rapidly advancing pre-trained models, the scientific value and practical significance of ***carefully designing high-level decision frameworks*** to unleash the potential of these models entirely are ***equally important*** as developing ***new network architectures.*** We ***kindly request*** you to ***re-evaluate*** the contribution of our work from the perspective of it being an ***"innovative method addressing a core problem".***
>
> ## Weakness 2
> Our ***"self-adaptation"*** means that our strategy can ***dynamically adjust the direction of robot exploration*** based on real-time environmental feedback to achieve ***more purposeful exploration*** of the environment. The ***relevant support or evidence*** is the ***four triggers we introduced in 3.4.4.***
>
> ***The main distinction from previous graph-based methods***：
> - In the future version, the distinction from previous graph-based methods will be made more meticulously, and thank you for this insightful comment. The key distinction lies in the objective: while previous methods primarily focus on modeling ***spatial topology [1,2,3 ...]*** or ***building scene graphs [4,5 ...]*** for semantic relationship representation, ***our Region-junction Graph*** is specifically designed to extract and ***represent independent semantic regions (e.g., bedrooms, living rooms)*** as fundamental units for exploration.
>
> refs
> >
> 1. VoroNav: Voronoi-based Zero-shot Object Navigation with Large Language Model ICML 2024
> 2. Beyond the nav-graph: Vision-and-language navigation in continuous environments. ECCV 2020
> 3. Neural topological slam for visual navigation. CVPR 2020
> 4. SG-Nav: Online 3D Scene Graph Prompting for LLM-based Zero-shot Object Navigation. NeurIPS 2024.
> 5. UniGoal: Towards Universal Zero-shot Goal-oriented Navigation. CVPR 2025.
> >
>
>
> ## Weakness 3
> ***Firstly,*** the proposed ***Dynamic Policy Trigger (DPT)*** allows the agent to dynamically adjust the target region node according to the environment perception feedback, which is to ***resolve the first issue***.
>
> ***Secondly***, by constructing the ***Region-Junction Graph*** to model the environment and ***treating independent functional regions*** (like bedrooms, living rooms) as ***exploration unit***, we effectively ***address the second issue.***
>
> Furthermore, I would like to take this opportunity to gently clarify a point in your inquiry.
> You mentioned:
> >"For example, does the DPT module update frontiers at varying time steps? Do we only consider one frontier in each semantic region, or consider all frontiers in the semantic region as a whole for the decision?"
> >
>
> The key point here is that our DPT module could ***updates "Region Nodes" at varying time steps*** according to the environment perception feedback, ***not "frontiers".*** This is because our approach ***entirely moves away from the FBE strategy;*** the fundamental ***unit for exploration*** is the ***semantic region itself, not traditional frontiers.***

---

> ### Author Response · Authors · 2025-11-21
> **Response for Weakness 4,5**
>
> ## Weakness 4
> You raise a very important point.  It is true that we ***initially identify Region Nodes using a purely geometry-based method***, which ***does not involve semantics*** at this stage.  ***However,*** as soon as a ***Region Node is recognized from the relevant RGB information***, we promptly ***update its Representative Perspectives (RP)***.  The updated RP then incorporates both ***visual and semantic information*** about the region.  ***Subsequently***, the ***VLM analyzes the RP*** of each Region Node to assess the ***likelihood of the target being present in that region.***
>
> ## Weakness 5
> We sincerely appreciate your feedback regarding the presentation of the four triggers in Section 3.4.4.
>
> We think that we have provided a detailed description of the triggering conditions for each trigger in the submission. ***Could you be able to specify which parts remain unclear？***, we would be very grateful and can offer further clarification.
>
> The design of these triggers is ***indeed closely linked to our Region-junction graph***, which reflects the ***coherence and rigor of our method***.
>
> We believe the core idea of ***utilizing triggers for dynamic exploration*** can be ***adapted or borrowed*** by other researchers and integrated with ***diverse environmental representations*** to enable ***flexible policy adjustments or other requirements.***

---

> ### Author Response · Authors · 2025-11-21
> **Response for Question 1,2**
>
> ## Question 1
>
> Thank you for raising this point. The rationale behind the ***radius threshold*** is that a ***junction***, which represents a ***location providing access to multiple regions***, might be detected at ***slightly different positions across time steps*** due to ***sensor or estimation noise.***
>
> The ***radius threshold*** is essential for ***correctly merging these observations of the same physical junction into a single node***. It cannot be removed because, ***without it,*** the graph would ***contain multiple redundant nodes for a single physical junction***, compromising the graph's ***accuracy.***
>
> Conversely, ***distinct junctions*** that are far apart should ***remain separate.*** Therefore, the threshold is critical for distinguishing between repeated detections of the same junction and different junctions that should be kept distinct.
>
> ## Question 2
> The differentiation between the ***Progressing List*** and the ***Backtracking List*** is primarily designed to address a ***potential misinterpretation*** that could occur if we relied ***solely on the preference score (PS) for sorting.*** The PS combines the ***VLM score (positively correlated) ***and the ***distance score (negatively correlated),*** meaning a*** higher VLM score*** and a ***closer distance*** both result in a ***higher PS.***
>
> ***The closer an agent is to a region, the richer visual information it obtains.*** If the ***VLM score remains low*** under such ***close observation*** (indicating a ***low probability of the target's presence***), then the ***Preference Score (PS)*** for this region may still be ***inflated*** due to the ***short distance.*** Sorting regions ***purely by PS*** in descending order could therefore ***misprioritize these low-potential areas.***
>
> To mitigate this, we categorize regions into two lists:
> - The ***Progressing List*** contains general candidate nodes for ***primary exploration.***
>
> - The ***Backtracking List*** specifically holds regions that are close in distance but have low VLM scores, suggesting they are low possibility of containing the target after detailed scrutiny.
>
> The agent ***first ***explores all nodes in the ***Progressing List***, and only considers the Backtracking List afterward. This strategy is supported by our experimental results: in all successful episodes, ***94.5%*** of targets were found during the ***Progressing phase***, while only*** 5.5%*** were found in the Backtracking phase. Meanwhile, our method has also achieved a ***higher success rate*** and ***path efficiency***. This validates that the rationality and effectiveness of our distinction between ***Progressing List*** and ***Backtracking List***.

---

> ### Author Response · Authors · 2025-11-21
> **Discussion**
>
> Dear reviewer ZTxk, we have done our best to address all of your concerns and questions. Should you have any further comments, we would be very happy to discuss them with you.

---

### Official Review · Reviewer_Jzgx · 2025-11-01

**Soundness:** 2
**Presentation:** 2
**Contribution:** 2
**Rating:** 2
**Confidence:** 4

**Summary:**

This paper proposes DP-Nav, which performs VLM-guided dynamic exploration on a Region–Junction Graph (RJG). The system comprises four modules: (1) RJG construction from RGB-D by extracting traversable skeletons and identifying region/junction nodes; (2) Representative Perspectives (RP) that store multi-view evidence per region; (3) a Scoring–Screening Mechanism (SSM) that fuses a VLM score and distance-based path cost to split nodes into a Progressing List (PL) and a Backtracking List (BL); (4) a Dynamic Policy Trigger (DPT) with four events (region discovery, perspective update, junction pass, region reached) to activate SSM for on-demand replanning. Experiments on ObjectNav, TextNav and InstanceNav over Gibson/HM3D/MP3D report improvements in SR/SPL.

**Strengths:**

- The decision unit is lifted from frontier points to the region level.
- Figures are visually appealing and clearly convey the main ideas.
- The writing is clear and easy to follow.

**Weaknesses:**

- Although the specific implementation details differ slightly, the overall idea appears quite similar to existing approaches that also rely on region-based potential estimation and multi-scale scoring mechanisms. Without a clearer articulation of the conceptual innovation or the insight that motivates this design, the contribution feels incremental.
- Heavy hand-tuning of triggers and thresholds makes the pipeline look like a “strategy stack”; the learning-based adaptivity appears limited. Sec. 3.4.4 defines four trigger types, specifies update rules via formulas, and fixes a trigger priority order—these timings and conflict resolutions are hand-crafted. In SSM, the fusion weights/thresholds (e.g., the near-distance and low-score thresholds and diversity parameters in Eqs. 7–11) are also manually set, with little justification or comparative evidence as to why these choices are necessary.
- RJG construction relies on skeletonization with depth thresholds, traversability masks, and node degree. It is unclear whether this geometric heuristic is stable in challenging cases such as cross-level transitions, stairs, or irregular rooms, and whether it can properly merge multiple entrances of the same room or handle mixed room types when identifying nodes.
- The region potential mainly comes from the VLM score of RP plus a distance term, with no explicit semantic support or updatable pixel-level evidence. Under viewpoint changes, occlusion, or similar-appearance distractors, the score reliability is uncertain. Although the authors use Progressing and Backtracking lists, there seems to be no record of “sufficiently searched but goal not found”, which may still cause repeated exploration of already visited regions.

**Questions:**

- After reaching a region, how do you define “sufficiently searched so we won’t return”?
- What is the rationale for the fixed priority order of the four triggers? In practice, what are the frequency histograms and average intervals of each trigger?

---

> ### Author Response · Authors · 2025-11-21
> **Response for Weakness 1**
>
> ## Weakness 1
> Thank you for this observation. You noted that
> >"the overall idea appears quite similar to existing approaches that also rely on region-based potential estimation and multi-scale scoring mechanisms."
> >
>
> To ensure we address your concern accurately and to better understand the specific works you have in mind, could you please suggest ***some relevant paper references?*** This would allow us to provide a more targeted and substantive comparison in our response.
>
> Furthermore, we would like to gently emphasize that the ***core contribution*** of our work lies in the novel integration of a ***region-junction graph with a four-trigger dynamic policy mechanism***. To the best of our knowledge, and based on a comprehensive review of the literature, this represents the ***first effort*** in this field to systematically ***address the limitations of static navigation policies through a dynamic exploration strategy.***
>
> We would like to clarify our core contributions from two key aspects, building upon the original submission:
>
> - ***1. Region-based Exploration:*** Previous methods often rely on frontier-based exploration, which may ***fragment a coherent semantic region*** (e.g., a single bedroom) into ***multiple sub-parts.*** For instance, one bedroom might contain three frontiers, requiring the agent to explore it ***three separate times—without guarantee of consecutive visits***. In contrast, ***DP-Nav*** models the environment using a ***Region-Junction Graph***（Each ***region node*** represents an independent ***semantic area, such as a bedroom or a toilet)*** and conducts exploration at the semantic region level. This ensures that ***each region***, such as a bedroom, is explored thoroughly in a ***single visit***, significantly ***reducing path redundancy***.
>
> - ***2. Dynamic Exploration:*** Prior approaches typically update long-term goals at ***fixed intervals (e.g., every 25 steps)***. A key limitation is that the agent may ***overlook potential opportunities***—such as a bedroom or a living room when finding a chair—encountered along the way to its current long-term goal. If the agent can ***dynamically pause its primary goal*** and take a few steps to ***investigate***, it can ***avoid the need for a costly return trip later***. This dynamic adjustment mechanism allows for ***more efficient use of the limited 500-step budget***, minimizing unnecessary backtracking and ***improving overall navigation efficiency.***
>
> So we ***don't*** think our innovation is ***incremental*** because our focus is on ***dynamic policies***, which were ***not paid attention to in previous work like [1,2,3].*** Finally, we sincerely hope the above can solve your concerns.
>
> Refs
> >
> 1. SG-Nav: Online 3D Scene Graph Prompting for LLM-based Zero-shot Object Navigation. NeurIPS 2024.
> 2. UniGoal: Towards Universal Zero-shot Goal-oriented Navigation. CVPR 2025.
> 3. GAMap: Zero-Shot Object Goal Navigation with Multi-Scale Geometric-Affordance Guidance. NeurIPS 2024.
> >

---

> ### Author Response · Authors · 2025-11-21
> **Response for Weakness 2**
>
> ## Weakness 2
> Our pipeline is indeed based on***carefully designed triggers,*** which stems from our objective of developing ***a zero-shot, training-free framework***. In such a setting, the adaptability is achieved not through learning from scratch, but through ***principled rules*** that leverage the ***semantic reasoning capabilities of pre-trained foundation models (like VLMs)***. We acknowledge that this ***differs from a learning-based approach; however, we posit that it represents a different but equally valuable paradigm***. This design choice allows ***DP-Nav*** to be highly efficient, transparent, and easily reproducible ***without the need for extensive computational resources for training***, which we believe is a significant advantage for practical applications and for establishing a robust baseline in the field.
>
> About the Four Triggers :
> - **(1)** We would like to clarify that the activation of the ***four triggers*** is ***inherently random***, as each trigger operates independently based on its own set of conditions ***without a fixed sequence***. The term ***"priority"*** mentioned in our paper specifically refers to the ***conflict-resolution*** mechanism used when the conditions for ***multiple triggers are met simultaneously***. In such cases, the system executes the trigger with the ***highest predefined priority*** to ensure ***decision-making consistency***.
> - **(2)** We fully agree with you that these components are ***hand-crafted in nature***. However, within the context of ***a zero-shot framework***—which by definition operates ***without task-specific training***—such design choices are not only inevitable but also integral to the pipeline's functionality.
> - **(3)** The  ***absence of learning-based adaptation*** is a ***deliberate characteristic*** of our approach, which seeks to achieve intelligent behavior through the ***principled integration of pre-trained models*** rather than through ***additional training***.
>
> ***The hyperparameters in Eqs 7-11：***
> - ***hyperparameters in Eq 8:*** Our experiments indicate a ***positive correlation ***between the value of*** K*** (the number of Representative Perspectives per region node in Eq. 8) and ***navigation performance***. This is because more visual information helps VLM to analyze more accurately the possibility of a target existing in a certain region. However, a ***larger K*** requires the VLM to process more visual information per node, significantly ***increasing inference time and computational cost***. We found that  ***K>=7***, the ***performance improvement diminishes considerably*** , while the **computational cost continues to rise sharply**. Therefore, **K=6** was selected as it offers an ***optimal balance between high performance and manageable cost.***
>
> - ***hyperparameters in Eq 9:*** The determination of these two hyperparameters  ***φ and ζ*** in Equation 9  was obtained through a large number of experiments. The table below shows the SR(%) and SPL(%) under different parameter combinations. Our goal is to ***maximize the SR and SPL.***
>
> | φ \\ ζ       | 0.2         | 0.3         | 0.4         | 0.5         | 0.6         |
> |:-------------:|:-------------:|:-------------:|:-------------:|:-------------:|:-------------:|
> | 1            | 56.2 - 29.1 | 58.8 - 31.2 | 59.5 - 32.8 | 56.3 - 29.4 | 55.0 - 28.6 |
> | 2            | 58.5 - 30.8 | 56.9 - 30.5 | 61.0 - 33.9 | 58.7 - 31.2 | 58.1 - 29.5 |
> | 3            | 60.2 - 31.5 | 60.0 - 31.2 | **62.5 - 35.6** | 59.8 - 31.1 | 59.3 - 30.8 |
> | 4            | 59.8 - 30.9 | 61.5 - 32.1 | 60.9 - 33.5 | 58.4 - 30.3 | 56.7 - 29.2 |
> | 5            | 57.5 - 27.7 | 58.3 - 31.0 | 58.1 - 31.4 | 56.0 - 28.5 | 54.2 - 27.1 |
>
> -  ***hyperparameters in Eq 11:*** The VLM score is obtained by VLM after thoroughly analyzing the visual information of the candidate regions. We believe that the VLM score is more important for determining whether a region is worth exploring. So, based on the inspiration from previous work in the formula, we determined that γ being 0.6(Really manually set) in Eq. 11 indicates that this evaluation of regional nodes will pay more attention to semantic information. And λ, as the distance decay coefficient, was commonly used as a fixed hyperparameter in previous works [1,2] about robotics and path planning.
>
> - ***It should be noted that*** for the optimization of hyperparameters, we only optimize based on the training scenario of Object goal navigation of HM3D. Then perform verification directly on the Gibson and MP3D datasets as well as Text Goal Navigation and Instance Image Goal Navigation, without the need to readjust the hyperparameters on the new datasets and new tasks.
>
> refs
> >
> 1. Enhancing Multi-Robot Semantic Navigation Through Multimodal Chain-of-Thought Score Collaboration. AAAI 2025
> 2. ApexNav: An Adaptive Exploration Strategy for Zero-Shot Object Navigation with Target-centric Semantic Fusion. RAL 2025
> >

---

> ### Author Response · Authors · 2025-11-21
> **Response for Weakness 3,4**
>
> ## Weakness 3
>
> In our approach, the depth map obtained from the RGB-D camera represents the distance from each pixel in the RGB image to the camera. Using Equations (1) and (2), we convert this distance into the vertical height of each pixel relative to the camera. Pixels with smaller vertical distances are considered to belong to objects that are higher off the ground and are thus treated as obstacles. Conversely, pixels with larger vertical distances are more likely to belong to the ground plane. Since the robot navigates on a flat floor, we empirically determined the hyperparameters in Equation (1) (as described in Lines 199-207 of the manuscript) to generate a traversability mask for each viewpoint.
>
> We acknowledge that this method is ***primarily designed for single-floor indoor environments*** and ***may not*** generalize well to scenarios involving ***multiple floors or complex terrains.*** Furthermore, ***navigation in multi-floor settings*** represents a ***distinct research direction***, as explored in other works such as [1, 2, 3, 4]
>
> refs
> >
> 1. Development of an indoor delivery mobile robot for a multi-floor environment. IEEE Access 2024
> 2. Nv-liom: Lidar-inertial odometry and mapping using normal vectors towards robust slam in multifloor environments. IRAL 2024
> 3. Multi-floor zero-shot object navigation policy. ICRA 2025
> 4. Stairway to Success: An Online Floor-Aware Zero-Shot Object-Goal Navigation Framework via LLM-Driven Coarse-to-Fine Exploration. arXiv 2025
> >
>
> For
> >"whether it can properly merge multiple entrances of the same room"
> >
>
> Please allow us to illustrate with an example. For example, if one bedroom has two doors, our method will recognize both doors as entrances to that bedroom. It will not merge the two physical doors into one. This ensures that the constructed ***Region-Junction Graph*** remains ***consistent with the physical conditions.***
>
> For
> >"whether it can handle mixed room types when identifying nodes."
> >
>
> We would like to clarify that our region node identification method is fundamentally ***geometry-based***. As long as a ***semantic region*** has an ***accessible opening***, our algorithm can ***recognize it***. This approach is ***independent*** of whether the ***room types*** are ***mixed or single***, as it focuses on the geometric connectivity of the space rather than semantic classification of room functionality. The specific ***semantic information*** about this region node will be analyzed by the ***subsequent VLM.***
>
> ## Weakness 4
>
> In response, we would like to clarify that the VLM score for each candidate region is indeed derived by the VLM based on visual information(Representative Perspective). The ***Representative Perspective*** has already provided ***rich semantic information for the VLM***. Furthermore, as detailed in ***lines 253-268 of our manuscript***, we specifically address the issue of reliability by ***capturing multiple Representative Perspectives*** for each region(And we conducted ***experiments on Representative Perspectives in Table 2 of original submission***). This multi-view strategy is designed to mitigate the ***uncertainties caused by factors like occlusion or partial views***, thereby ***increasing the confidence and stability of the VLM's semantic assessment.***
>
> The ***rest of  Weakness 4*** here are ***closely related to your Question 1***, and we have provided a detailed response there. Please ***refer to*** our response to ***Question 1*** for clarification.

---

> ### Author Response · Authors · 2025-11-21
> **Response for Question 1,2**
>
> ## Question1
> Upon reaching a target region node, our agent comprehensively collects RGB information from the current region. These visual observations are systematically organized according to the agent's viewing perspective from left to right and assigned sequential identifiers. The indexed RGBs are then processed by the VLM to evaluate the likelihood of the target object's presence in the current region. For instance, when searching for a bed, if the VLM determines the region to be a living room based on semantic understanding of corresponding Representative Perspectives , the agent terminates exploration in this region immediately and excludes this region from future revisits. The SSM subsequently assigns a new target region node for continued exploration. Conversely, if the VLM identifies a potential target presence, it returns the identifier of the perspective most likely to find the target.  The agent then proceeds to investigate in that specific direction.
>
> ***For implementation details, please refer to Section A.4 of our original manuscript.***
>
> ## Question 2
> This Question  here is ***closely related to Weakness 2***, and we have provided a detailed response there. Please ***refer to*** our response to ***Weakness 2*** for clarification. Furthermore, We analyzed the experimental data to calculate the average trigger interval and frequency of the four triggers.
>
> | Dynamic Policy Trigger  |Average Activation Interval (Timesteps)|Average Frequency (Times)|
> | :---:                                |   :-----:  |         :---: |
> |Region-Reached Trigger  | 48       | 5   |
> |Region-Discovery Trigger| 29        | 8      |
> | Perspective-Update Trigger| 20        | 12      |
> | Junction-Pass Trigger | 51        | 6      |

---

> ### Author Response · Authors · 2025-11-21
> **Discussion**
>
> Dear reviewer Jzgx, we have done our best to address all of your concerns and questions. Should you have any further comments, we would be very happy to discuss them with you.

---

### Official Review · Reviewer_57Qp · 2025-11-01

**Soundness:** 3
**Presentation:** 3
**Contribution:** 3
**Rating:** 8
**Confidence:** 3

**Summary:**

This paper introduces a dynamic framework based on semantic regions, called DP-Nav to enable semantic navigation in embodied agents. This is in contrast to existing frontier based methods that ignore semantic region information. DP-Nav builds an additional region-junction graph to keep track of semantic regions and the corresponding frontiers. Through experiments the authors show that their method outperforms SOTA methods.

**Strengths:**

The idea to keep track of semantic regions seems to be useful to efficiently explore the environment. The paper is well written and easy to understand.

**Weaknesses:**

I find the work presented in this paper to provide useful insights for the community. I didn’t find any significant weaknesses.

**Questions:**

1. Do you think the region potential can be extended to incorporate textual semantics (e.g., “bathroom” vs. “corridor”)?

Minor comments:
The formatting of some table captions needs to be revised. For example, it’s hard to distinguish the wrapped text from the caption of Table 4. Same for Figure 3.

---

> ### Author Response · Authors · 2025-11-21
> **Response for Question 1**
>
> # Questions 1
> Thank you for this valuable suggestion. In our current implementation, the **region potential** is assessed by combining the VLM's evaluation of its **Representative Perspectives** with the **agent's distance to that region**. Indeed, the **VLM can already** infer semantic room types (e.g., "bathroom" or "corridor") from visual information. We agree that explicitly incorporating such textual semantics would be particularly **beneficial** when visual information is **incomplete or occluded**. In such cases, the textual cues can better assist and guide the VLM in analyzing the ambiguous visual content. However, integrating this directly would require a new module to pre-identify room types like "bathroom" or "corridor". As we must **remain faithful to our original submission** during rebuttal, such an extension is not feasible at this stage. We will further explore this promising scheme in our future work. We sincerely appreciate this insightful idea and will definitely consider it as an important direction for our future work.
>
> # Response to the Minor comments:
> Thank you for carefully reading our paper. In the future version, we will make **adjustments** to **Table 4 **and** Figure 3** and carefully **examine** the **other charts**.

---

> ### Author Response · Authors · 2025-11-21
> **Discussion**
>
> Dear reviewer 57Qp, we have done our best to address all of your concerns and questions. Should you have any further comments, we would be very happy to discuss them with you.

---

### Meta-Review · Area_Chair_osFG · 2026-01-06

**Summary:**

This paper introduces DP-Nav, a dynamic exploration framework for zero-shot VLN that treats semantic regions as policy units. It builds a Region–Junction Graph (RJG) and uses a Scoring–Screening Mechanism (SSM) together with Dynamic Policy Triggers (DPT) to re-plan exploration targets on demand. Reviews were mixed: one reviewer found the approach useful and did not identify major weaknesses, while two reviewers raised strong concerns that the core ideas and design choices are largely incremental and engineering-oriented rather than offering a clearly distinguishable conceptual advance over closely related region/graph-based exploration pipelines.

**Reviewer Concerns:**

During rebuttal, the authors provided detailed clarifications on (i) how regions are terminated/excluded from revisits after VLM-based assessment, (ii) the intended meaning of “dynamic/self-adaptive” behavior via DPT, including trigger statistics, and (iii) hyperparameter choices and claims of cross-task/cross-dataset transfer without retuning. However, the primary decision-driving concern which raised by multiple reviewers, was whether the paper makes a sufficiently contribution beyond a system-level integration of known components. While the rebuttal improved clarity, in the view of the ACs, it did not adequately address the reviewers’ concerns regarding novelty and insight to alter the overall evaluation.

**Reviewer Scores:**

Initial ratings: 57Qp=8, Jzgx=2, ZTxk=2, 8dHA=4; Updates: none stated. No post-rebuttal score updates were recorded. The ACs recommend to reject this paper.

---

### Decision · Program_Chairs · 2026-01-26

Reject